# Modeling Microscale Foam Propagation in a Heterogeneous Grain-Based Pore Network with the Pore-Filling Event Network Method

Jun Yang [1,2], Nu Lu [3], Zeyu Lin [2,4], Bo Zhang [5], Yizhong Zhang [6], Yanfeng He [1] and Jing Zhao [1,*]

1 School of Petroleum and Natural Gas Engineering, Changzhou University, Changzhou 213164, China
2 Faculty of Engineering and Applied Science, University of Regina, Regina, SK S4S 0A2, Canada
3 Research Institute of Petroleum Exploration and Development, PetroChina, Beijing 100083, China
4 Oil Production Research Institution, Dagang Oilfield of China National Petroleum Corporation (CNPC) Ltd., Tianjin 300280, China
5 CNPC Research Institute of Safety and Environment Technology, Beijing 102206, China
6 Cooperative Innovation Center of Unconventional Oil and Gas, Yangtze University, Wuhan 430100, China
* Correspondence: maggiezhao@cczu.edu.cn

**Abstract:** Foam flooding is an efficient and promising technology of enhanced oil recovery that significantly improves sweep efficiency of immiscible displacement processes by providing favorable mobility control on displacing fluids. Although the advantages in flexibility and efficiency are apparent, accurate prediction and effective control of foam flooding in field applications are still difficult to achieve due to the complexity in multiphase interactions. Also, conventional field-scale or mesoscale foam models are inadequate to simulate recent experimental findings in feasibility of foam injection in tight reservoirs. Microscale modeling of foam behavior has been applied to further connect those pore-scale interactions and mesoscale multiphase properties such as foam texture and the relative permeability of foam banks. Modification on a microscale foam model based on a pore-filling event network method is proposed to simulate its propagation in grain-based pore networks with varying degrees of heterogeneity. The impacts of foam injection strategy and oil-weakening phenomena are successfully incorporated. Corresponding microfluidic experiments are performed to validate the simulation results in dynamic displacement pattern as well as interfacial configuration. The proposed modeling method of foam propagation in grain-based networks successfully captures the effects of lamellae configurations corresponding to various foaming processes. The results of the simulation suggest that the wettability of rock has an impact on the relevance between reservoir heterogeneity and the formation of immobile foam banks, which supports the core idea of the recently proposed foam injection strategy in tight oil reservoirs with severe heterogeneity, that of focusing more on the IFT adjustment ability of foam, instead of arbitrarily pursuing high-quality strong foam restricted by permeability constraints.

**Keywords:** foam flooding; pore network modeling; invasion percolation; enhanced oil recovery

## 1. Introduction

Benefitting from its effectiveness in improving sweep efficiency and optimizing mobility adjustment, foam flooding has been developed for decades in field application of enhanced oil recovery (EOR), especially for reservoirs with severe heterogeneity and poor injectivity of other chemicals like polymer [1–3]. In the foam flooding application, foaming gas and foaming surfactant solution are injected to generate foam in the reservoir. The presence of foam can significantly reduce the mobility of displacing fluids and provide favorable profile control, which eventually improves the sweep efficiency of the immiscible displacement process [4–6].

Like other EOR methods, improving the effectiveness and optimization of the injection strategy is essential to further release the potential of foam flooding in field application.

Thus, the dependable prediction of foam propagation in porous media based on the dynamic interaction between multiphase fluids is the key to resolving current problems and limits in foam-related field applications [7–11].

To date, there are four types of common foam modeling methods according to the model establishment approaches, including the population balance model, semiempirical model, fractional flow model, and percolation-based model [12–15]. The population balance model is widely used and optimized for its advantages in easy implementation in multiple commercial or noncommercial reservoir simulators. However, due to the scope of modeling objectives, the lack of mechanistic details such as interfacial phenomena during foam propagation inside porous media becomes an issue in quantifying the model parameters such as foam texture, foam generation rate, and coalescence rate [16–19]. Meanwhile, a semiempirical model of foam flow in porous media is proposed to provide a rapid calculation that assists the field application of foam injection processes [20–22]. Poor generality has restrained further development like incorporating more mechanistic features [1,4,23]. The fractional flow model of foam flooding is modified from a similar tool used in water flooding processes to provide an efficient estimation of parameter selection of foam injection [24–26]. Currently, the relation between foam flow rate and the interfacial behavior of foaming activities cannot be appropriately incorporated, and results estimated by the fractional flow approach cannot cooperate with other foam simulators smoothly from different scales of objectives.

The percolation-based model provides a reliable quasistatic approach to incorporate many pore-scale mechanisms and interfacial activities into the algorithmic model which can easily be upscaled to mesoscale and even field scale in well patterns [14,27–29]. However, like other pore-scale modeling attempts, considerable work related to the efficiency of computation remains unresolved when new features keep being added to the framework. Additionally, current trapping identification based on a depth-first cluster labeling algorithm falls behind in both accuracy and efficiency.

Gradually, modeling foam propagation in porous media from the pore scale has become an effective option to explicitly incorporate interfacial behaviors into other numerical simulators to update the description of rock–fluid interactions of foaming processes [30,31]. Thus, by combining a percolation-based model of foam behavior and pore network modeling of immiscible displacement, a quasistatic pore-scale foam modeling approach is proposed and concluded as an invasion percolation with memory (IPM) algorithm, which extracts additional local resistance exerted from active foam lamellae as the memory term of foam flowing path [27,32]. Later, continuous studies were performed to further develop the foam model of IPM by adding post-breakthrough viscous effects, upscaling parameters, foam coalescence of capillary suction, etc. [29,30,33].

Recently, the IPM algorithm was incorporated into the framework of the pore-filling event network (PFEN) method to simulate foam propagation in grain-based pore networks, which allows a quantitative description of the interfacial configuration during the foam propagation with the varying wetting conditions [34]. However, there are several points that remain unresolved in previous work. First, the impact of heterogeneity on foam propagation inside grain-based pore networks has not been thoroughly discussed, which has direct effects on interfacial configurations. Second, the common application scenario, co-injecting gas and foaming surfactant into the porous media with the fixed foam quality, has not yet been realized in previous works of pore-scale foam models, which involves complicated multiphase interactions during injection. Third, the crucial oil-weakening effect on foam lamellae has also not been incorporated in previous studies [35].

Foam quality ($\Gamma$), which is defined as the volumetric gas fraction during foam injection, is one of the key parameters commonly used to control the in situ foam performance during the field application [36]. The difference in foam quality distinguishes flowing foam status into a strong foam regime with a higher gas fraction and a weak foam regime with a lower gas fraction [37–39]. Experimental studies have concluded that apparent viscosity is

independent of gas velocity inside the strong foam regime but becomes independent of gas velocity within the weak foam regime [40,41].

As the basic foam component in porous media, lamella is a thin liquid film located at the constricting pore space and making the gas phase discontinuous, as shown in Figure 1a. In oil-free scenarios, the lamella behavior is controlled by three types of generation mechanisms (snap off, leave behind, and division) and two types of coalescence mechanisms (capillary suction and gas diffusion) [30,42,43]. When local capillary forces become too great to be balanced by disjoining pressure exerted from the lamella itself, the filmlike lamella structure becomes unstable, continuously thinning until its rupture. As shown in Figure 1b, the presence of oil weakens existing lamellae and accelerates their rupture via four successive steps, from entering, spreading, bridging, to the rupture [44]. The occurrence of each event is the necessary precondition needed to activate the next event until the fulfillment of lamella rupture.

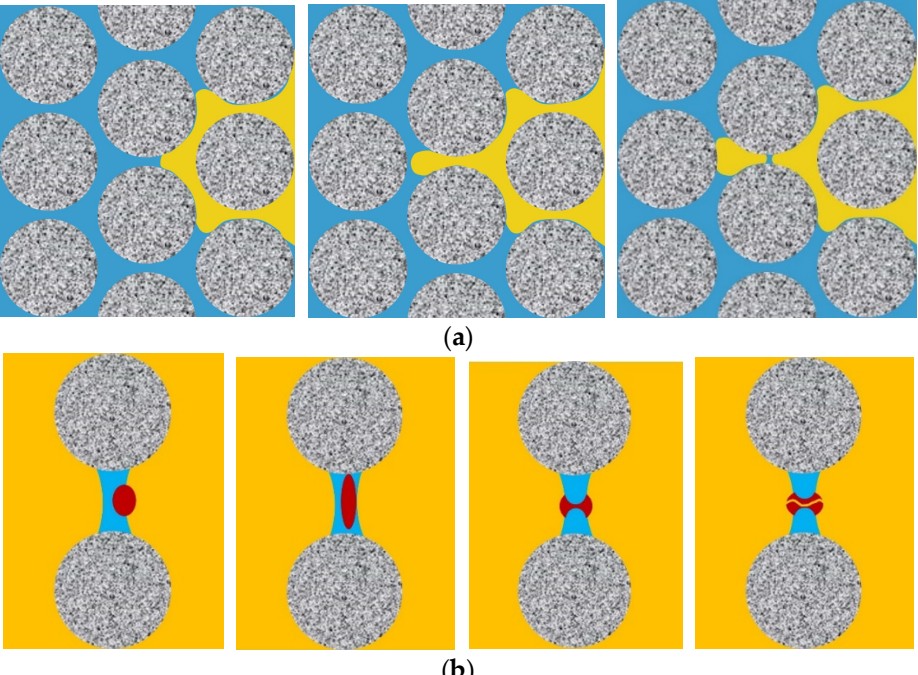

**Figure 1.** Schematic description of representative generation and coalescence of foam lamella. The gas phase is colored in yellow, the liquid phase is colored in blue, and the oil phase is red. (**a**) Typical generation of a foam lamella by snap-off; (**b**) Typical coalescence of foam lamella in the presence of oil, from entering, spreading, bridging, to the lamellae rupture [45].

In this work, a novel quasistatic modeling method is proposed to simulate foam propagation inside a heterogeneous grain-based pore network with the pore-filling event network (PFEN) method incorporated with oil-weakening effects, followed by a comparative investigation via microfluidic experiments.

## 2. Method

### 2.1. Category of Invasion Events

The core idea of modeling immiscible displacement with the PFEN method is converting the continuous invasion into a quasistatic process consisting of a series of invasion events defined by the pore-scale interfacial configuration. Cieplak and Robbins [46] proposed the method by quantifying and ranking the local pressure thresholds to mobilize frontal interfaces settling inside grain-based pore networks. As presented in Figure 2, the

$d_{OA}$, which is the distance between the center of grain circle A and the center of arc $A_c B_c$, can be calculated by Equation (1) [46],

$$d_{AO}{}^2 = r_A^2 + r_O^2 - 2r_A r_O \cos\theta \tag{1}$$

where $r_A$ is the radius of grain $A$, $r_O$ is the radius of curvature of the meniscus $A_C B_C$, and $\theta$ is the contact angle. With this relation, the local pressure threshold required to mobilize the meniscus can be solved in assist with the Young–Laplace equation of advancing curved plate, as shown in Equations (2) and (3) [46], where $\gamma$ is the interfacial tension between invading and defending fluids.

$$\frac{1}{r_O} = \frac{\Delta p}{\gamma} \tag{2}$$

$$\Delta p = \frac{\gamma}{r_O} \tag{3}$$

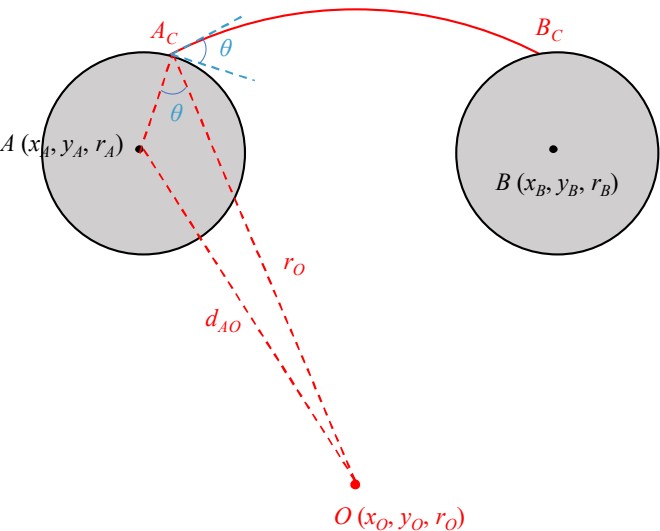

**Figure 2.** General interfacial configuration of a meniscus $A_C B_C$.

As presented in Figure 3a, three basic invasion events, including burst, touch, and overlap, are defined to cover the potential interfacial configuration in a general grain-based pore network which has a default coordination number of 3. The burst event, also known as the Haines jump, is the critical interfacial configuration sustaining the meniscus structure that any higher pressure will lead to the immediate invasion and filling of its target pore [47]. The threshold pressure of a burst event can be solved by combining the following equations with Equation (3) [46,48].

$$(x_O - x_A)^2 + (y_O - y_A)^2 = d_{AO}{}^2 = r_A^2 + r_O^2 - 2r_A r_O \cos\theta \tag{4}$$

$$(x_O - x_B)^2 + (y_O - y_B)^2 = d_{BO}^2 = r_B^2 + r_O^2 - 2r_B r_O \cos\theta \tag{5}$$

$$(y_B - y_A)x_O + (x_B - x_A)y_O - (y_B - y_A)x_A = y_O \tag{6}$$

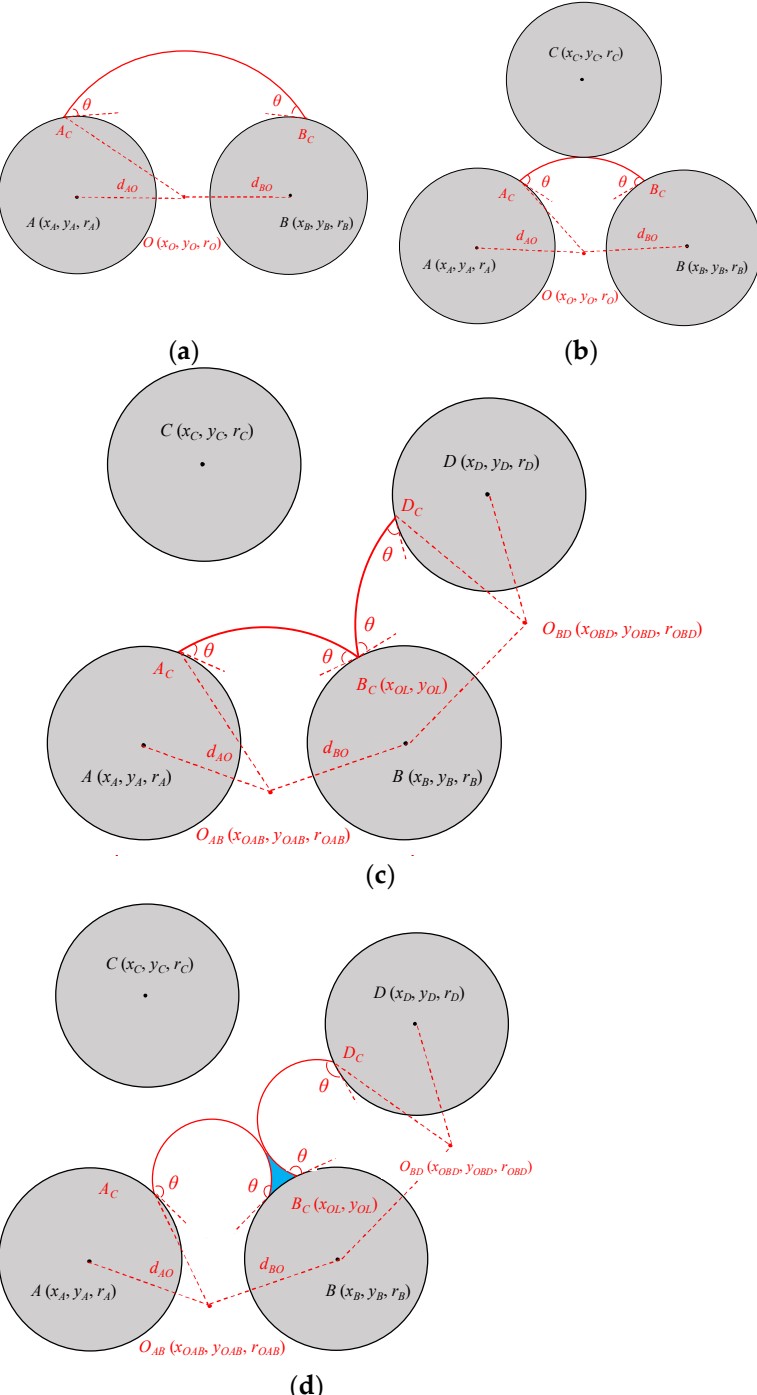

**Figure 3.** The interfacial configuration of the meniscus that triggers critical invasion events during immiscible displacement inside the grain-based pore network. (**a**) Burst; (**b**) Touch; (**c**) Overlap; (**d**) Drainage overlap.

Due to the compaction of grains that form the network, a meniscus may hit its target grain before fulfilling the local pressure threshold of burst, as shown in Figure 3b. Such an interfacial configuration is defined as touch, whose pressure threshold can be solved by combining the following equations and Equation (3) [46,48].

$$(x_O - x_A)^2 + (y_O - y_A)^2 = d_{AO}{}^2 = r_A^2 + r_O^2 - 2r_A r_O \cos\theta \tag{7}$$

$$(x_O - x_B)^2 + (y_O - y_B)^2 = d_{BO}^2 = r_B^2 + r_O^2 - 2r_B r_O \cos\theta \tag{8}$$

$$(x_O - x_C)^2 + (y_O - y_C)^2 = (r_O + r_C)^2 \tag{9}$$

Sometimes, two adjacent menisci may contact on their shared grain before fulfilling the pressure thresholds required to complete any invasion event, as shown in Figure 3c. Two menisci will merge when it occurs, whereas such an interfacial configuration is defined as overlap. The corresponding critical pressure of the overlap event can be solved by combining the following equations and Equation (3) [46,48].

$$(x_{OL} - x_{OAB})^2 + (y_{OL} - y_{OAB})^2 = r_O^2 \tag{10}$$

$$(x_{OL} - x_{OBD})^2 + (y_{OL} - y_{OBD})^2 = r_O^2 \tag{11}$$

$$(x_{OL} - x_B)^2 + (y_{OL} - y_B)^2 = r_B^2 \tag{12}$$

Recent works [34,48] stressed the specialization of the overlap event of the drainage process when the contact angle $\theta$ of the displacing phase is obtuse, which will result in the merge point detaching from the sharing grain B, as shown in Figure 3d. The distinction of drainage overlap indicates taking the presence of the residual defending phase into account, and the additional distance between the merging point and sharing grain should be added on the right-hand side of Equation (12).

### 2.2. Coupling of Foaming and Defoaming Events

Lamella is the basic unit of foam that intercepts continuous gas flow in porous media until the local pressure threshold is fulfilled. Conventional lamella generation and destruction mechanisms can be incorporated into a quasistatic displacement process of invading foaming gas by tracking the interfacial configuration among grain-based pore networks. Snap-off sites can be stochastically assigned on constriction points (representing pore throats) within the grain-based pore network based on a sitewise probability $p$. Specifically, a pore throat will be considered as a repetitive lamella generation site if its sitewise $p$ is not higher than $p_{SO}$, which is a global constant model parameter describing degree of foaming activity. A new foam lamella will be generated, as shown in Figure 4, when the continuous invading gas displaces the defending foaming liquid phase away from a pre-assigned snap-off site with $p < p_{SO}$.

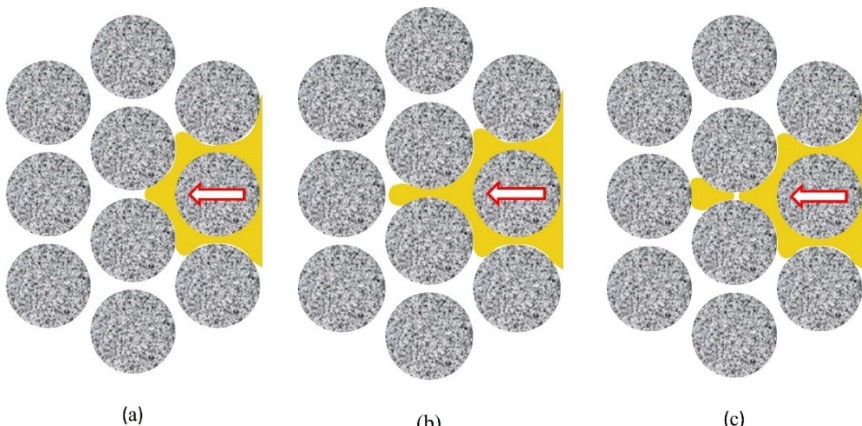

(a)　　　　　　　(b)　　　　　　　(c)

**Figure 4.** Schematic description of the lamella generation event triggered by snap-off inside a grain-based pore network. (**a**) The invasion front enters a pore throat; (**b**) The invasion front penetrates across the pore throat and stretches in a dumbbell shape; (**c**) A foam lamella generates due to the snap off of invading gas phase.

Defoaming events caused by coalescence of foam lamella is a complicated process involving surface thermodynamics and bubble dynamics. In porous media, foam lamellae are thermodynamically unstable due to the fluid exchange across the plateau border, exerting dynamic disjoining pressure to balance the local capillary forces by adjusting the thickness of the liquid film. When local capillary forces become too great to balance by self-thinning of static lamellae, as shown in Figure 5a, the lamellae structure becomes unstable and gradually thins until coalescence occurs. Recent studies [30,31] have developed a numerical approach to solve dynamic lamellae thickness based on the Derjaguin–Landau–Verwey–Overbeek (DLVO) method, which converts the local disjoining pressure isotherm to the relation between dynamic lamella thickness and time elapsed, as shown in Figure 5b, after the static lamella becomes unstable. Even though the lamellae are displaced away from the constricting pore throat with greater local capillary force when the pressure threshold is fulfilled, the potential squeezing and stretching during its transport make the mobilized lamella more vulnerable, as shown in Figure 6. During the mobilization of lamella inside pore space with periodical variation of pore structure, fluctuation of local geometric conditions requests lamella to adjust its thickness to match the capillary forces. In response, lamellae constantly squeeze and stretch during the transport by draining and refilling the liquid within, whereas the rupture occurs when refilling is slower than demands. Such an effect is simplified in the modeling of this work by raising the critical lamella thickness for transporting lamellae to qualitatively distinguish the difference between static and mobilizing films.

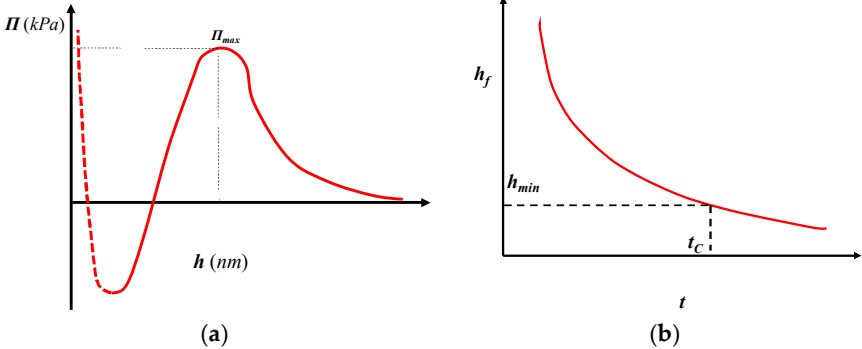

**Figure 5.** Schematic description of coalescence mechanisms of capillary suction. (**a**) Typical disjoining pressure isotherm of foam lamellae in porous media; (**b**) Relation of dynamic lamella thickness and time elapsed when the local capillary force is too great to balance.

In the field application of most foam injection projects, like foam flooding or foam-assisted $CO_2$ sequestration, foam is generated in situ by co-injecting or alternatively injecting gas and foaming liquid at a designated foam quality $\Gamma$, which is the gas fraction of both injected phases [4,42]. Also, the sequence of gas and liquid entering the porous media presents some degree of random character; even both fluids are injected at fixed foam quality $\Gamma$ during the co-injection foaming process [43,49]. Because the amount of fluid being introduced into the pore network at each invasion step is assumed to be constant for the pseudostatic process, which is exactly sufficient to fill one pair of pore body and pore throat, a stochastic model parameter ($p_i$) is introduced to assign the fluid type for the corresponding invasion step, either gas phase ($p_i \leq \Gamma$) or liquid phase ($p_i > \Gamma$). The workflow of stepwise-injected fluid corresponding to the entire process is shown in Figure 7, corresponding to the foam quality $\Gamma$ of co-injection workflow from 0.25, 0.5, to 0.75.

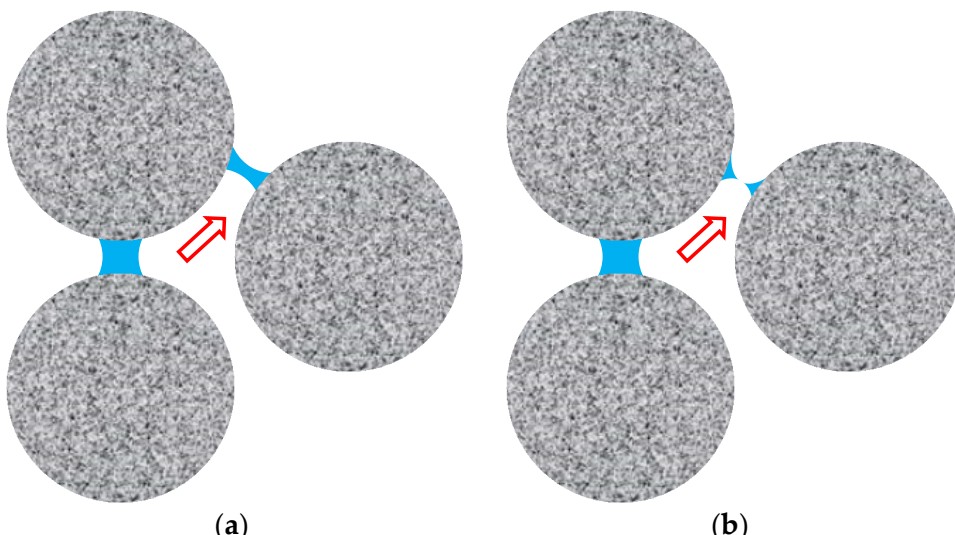

**Figure 6.** Transport of foam lamella causes additional fluctuation affecting its thickness. Red arrow represents the direction of the displacement. (**a**) Moderate fluctuation makes lamella thinner during its mobilization; (**b**) Lamella ruptures during mobilization with stronger fluctuation.

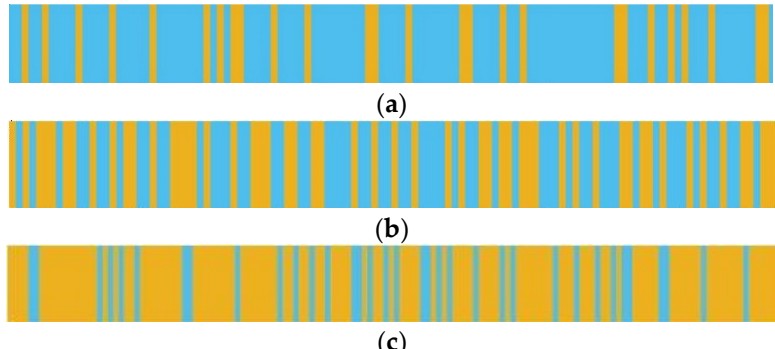

**Figure 7.** Schematic diagram of the workflow of injected fluid when the foam is co-injected at a different foam quality Γ. The liquid phase is blue; the gas phase is yellow: (**a**) Γ = 0.25; (**b**) Γ = 0.5; (**c**) Γ = 0.75.

Previous microscale experimental studies on foam propagation inside pore networks, as compared in Figure 8, also suggest potential variation of the foam regime although it is being co-injected at the same flow rate and foam quality. The mechanism of such a phenomenon is equivalent to the slug size control in the macroscopic surfactant-alternating-gas (SAG) process of foam injection since the real co-injection process is conflictive with the foundational algorithm of invasion percolation with memory. So, in order to compromise between the key model assumption and actual physics encountered during in situ foam generation, another model parameter, the minimum injected segment size $S_{min}$, is introduced to simulate the sparsification of in situ foaming phenomena, indicating the minimum invasion step number required before the next switch of injected fluid type, as shown in Figure 9. In this algorithmic model, $S_{min}$ works as an intervention time determining the fluid type. It is similar to the definition of slug size in field scale that lower $S_{min}$ indicates that fluid switch between gas and liquid is more frequent, as shown in Figure 9a, whereas higher $S_{min}$ will result in sparse foam configuration, as shown in Figure 9c, even though they have the same foam quality.

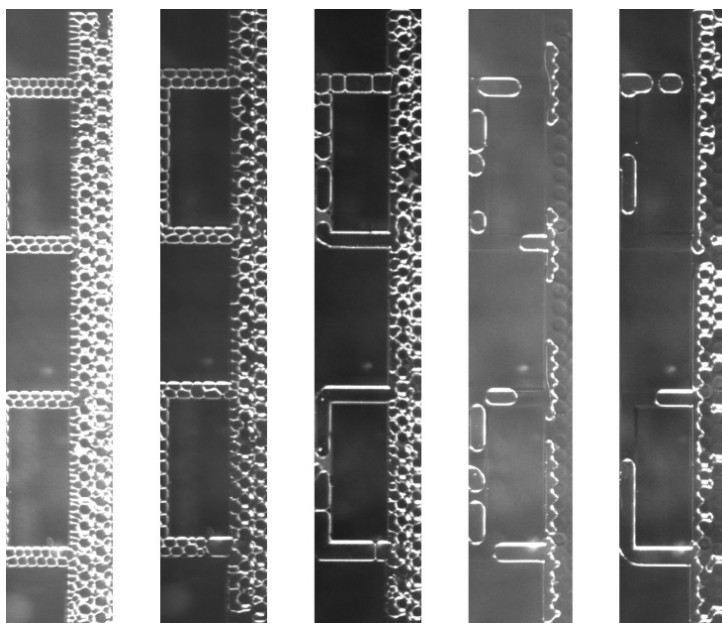

**Figure 8.** Despite foam quality and injection rate being controlled as the constant, the varying slug size of foam injection still results in a distinct flowing foam regime ($Q$ = 10 µL/min, $\Gamma$ = 0.8).

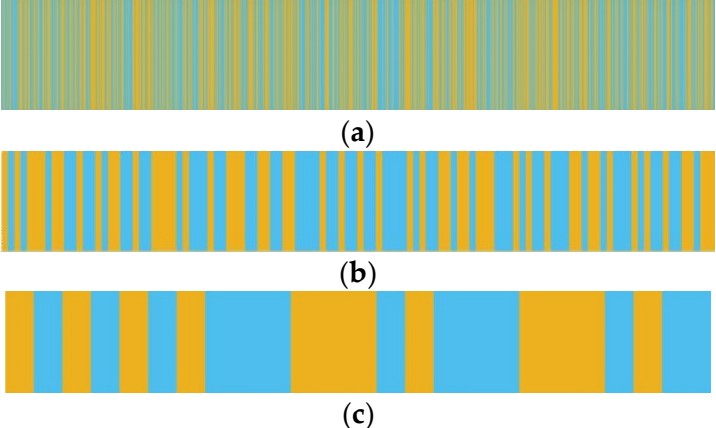

**Figure 9.** Schematic diagram of workflow of injected fluid when foam is co-injected at varying minimum injected segment size $S_{min}$. $\Gamma$ = 0.5. The liquid phase is blue; the gas phase is yellow: (**a**) $S_{min}$ = 1; (**b**) $S_{min}$ = 10; (**c**) $S_{min}$ = 50.

Previous pore-scale simulation attempts have developed over a decade to describe the oil-free drainage process of foaming gas being injected into the pore network saturated with foaming surfactant solution [27,30,33]. Foam lamellae generated during such a process temporarily block the pore throats and make gas flow discontinuous. The injected gas phase is distinguished as free gas and trapped gas as well. For the foam injection process in the presence of oil, interactions between the surfactant solution, oil, trapped gas, and free gas are important to adjust and quantify interfacial configuration during the simulation, as shown in Figure 10. Among these interactions, the oil-weakening effect on lamella is the key factor for the pore-scale simulation of the foam flooding process, which can be concluded as successive steps controlled by entering $E$ and spreading coefficient $S$, as shown in Figure 11.

Once the interfacial constraints of oil spreading within the lamellae are achieved, the impact of oil weakening on lamellae strength can be quantified with Equation (13) [30],

$$t_F = \int_{h_{FO}}^{h_F} -\frac{3F_D\mu_L R_F^2}{2(h_F)^3(p_{CA} - \Pi)}dh_F \tag{13}$$

where $\mu_L$ is the viscosity of foaming surfactant solution, $R_F$ is the equivalent radius of lamella structure, $p_{CA}$ is the local capillary pressure at the constricting part of the grain-based pore space, $t_F$ is the elapsed time of the invasion step, $h_{FO}$ is the lamella thickness at the beginning of the corresponding invasion step, $h_F$ is the dynamic film thickness of the foam lamella after $t_f$, and $F_D$ is the coefficient representing additional thinning effect due to oil ($F_D > 1$). $\Pi$ is the disjoining pressure exerted from the lamella.

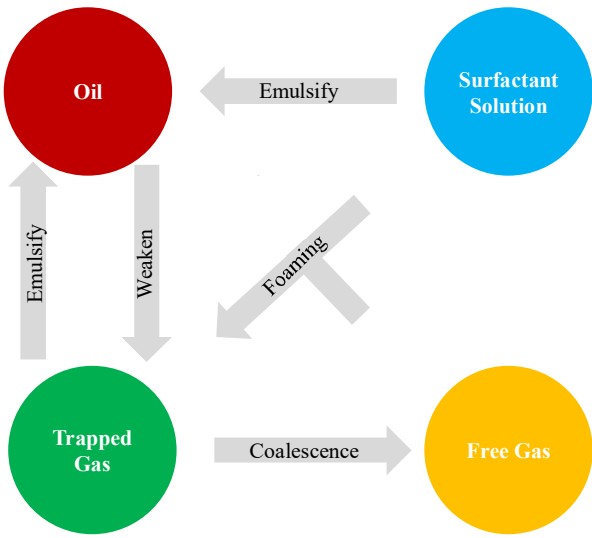

**Figure 10.** Key interactions during foam propagation in a pore network in the presence of the oleic phase.

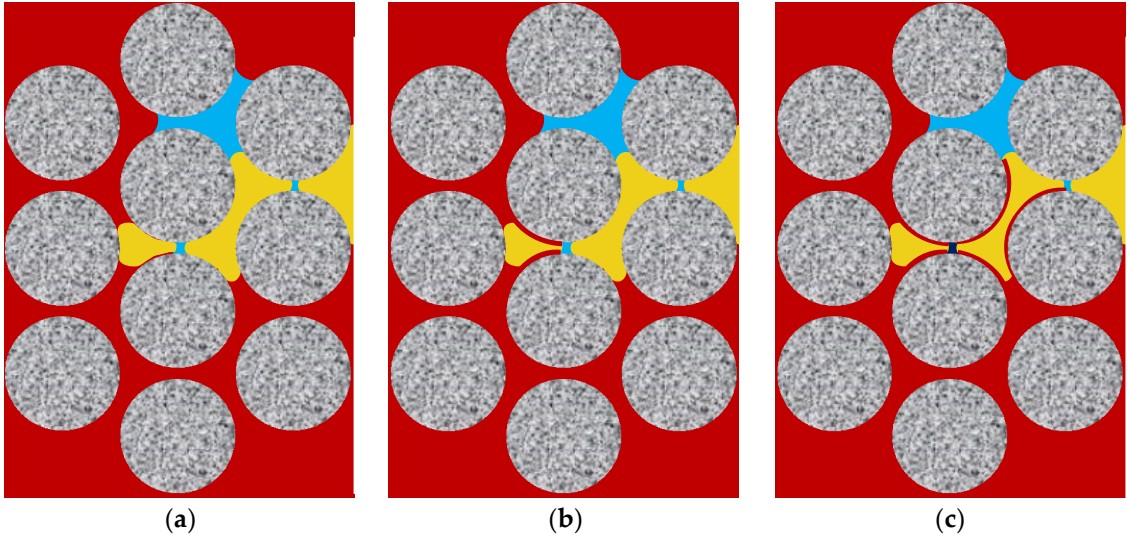

(**a**)  (**b**)  (**c**)

**Figure 11.** Pore-scale description of the oil-weakening effect on foam lamella. (**a**) $E \leq 0$, $S \leq 0$, oil does not enter the lamella; (**b**) $E > 0$, $S \leq 0$, oil enters the lamella but does not spread; (**c**) $E > 0$, $S > 0$, oil spreads inside the lamella.

### 2.3. Algorithm of Foam Propagation

The algorithm of foam propagation used in this work is developed by incorporating invasion percolation with memory (IPM) algorithm into the pore-filling event network approach. Similar to conventional immiscible displacement processes simulated based on the IPM algorithm [27,30,33], the invasion proceeds by identifying the frontal pore with the minimum pressure threshold $V$ as the sum of frontal capillary term and memory term. As the invasion scene shows in Figure 12, $j$ is the pore where the invasion front stands, $k_1$ and $k_2$ are two candidate pores of further displacement, and there is a foam lamella stand in the pore throat between $i$ and $j$. Equation (14) shows the pressure threshold of the potential invasion from pore $j$ to $k_1$ [31],

$$V(jk_1) = \xi(j) + p_{CE}(jk_1) \tag{14}$$

where $V(jk_1)$ is the total pressure threshold needed for the invasion, $\xi(j)$ is the memory term indicating the pressure threshold to mobilize the remaining active lamellae from $j$ to the inlet boundary along the involved foam flowing path, and $p_{CE}(jk_1)$ represents the capillary term as the pressure of the critical pore-filling event of meniscus $AD$. In this work, the memory term $\xi(j)$ can be estimated by Equation (15),

$$\xi(j) = [\Delta p_L(i,j) + \Delta p_L(g,i) + \Delta p_L(f,g) + \ldots] = \sum_{inlet}^{j} [\Delta p_L(x)] \tag{15}$$

where the pressure threshold of each active lamella combines the pressure thresholds of two pore-filling events, including the imbibition event of the leading meniscus and the drainage event of the trailer meniscus. After identifying the foam flowing path with the minimum pressure threshold, the invasion completes along the entire foam flowing path, whereas the condition and thinness will be updated accordingly, followed by a depth-first trapping identification throughout the entire network. The flowchart plotted in Figure 13 shows the procedure of modeling the foam propagation in the grain-based pore network with the pore-filling-event-based method. MATLAB R2021a is used in this study for simulation and plotting.

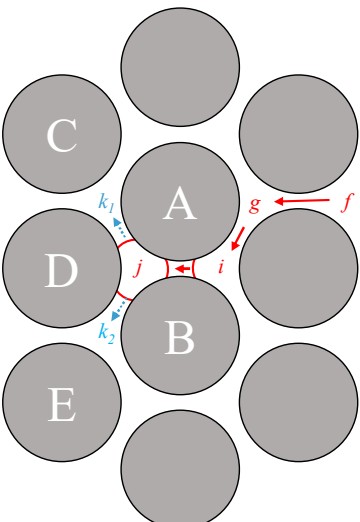

**Figure 12.** Schematic of typical interfacial configuration of invasion front.

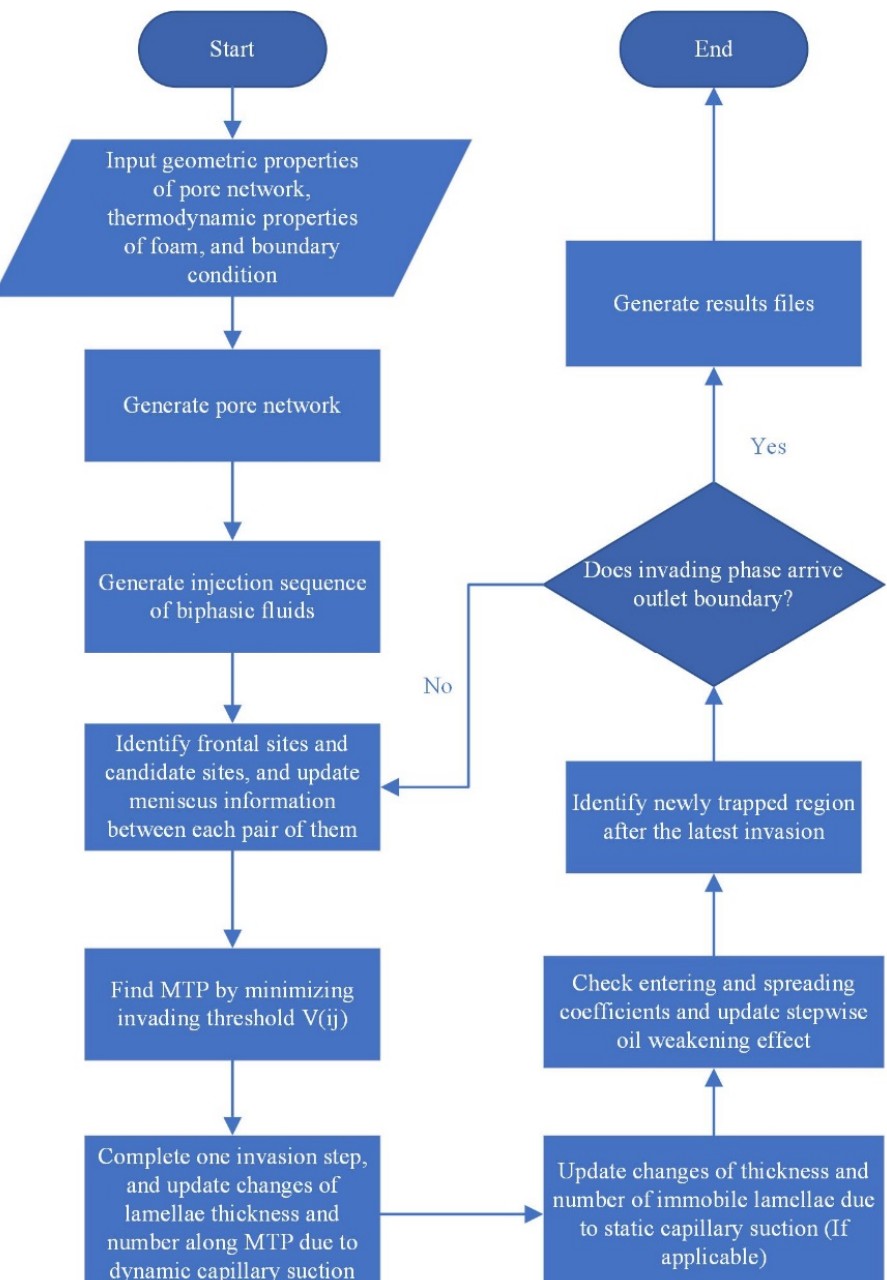

**Figure 13.** Flowchart of modeling foam propagation in a heterogeneous grain-based pore network with the pore-filling event network method.

### 3. Results and Discussion

*3.1. Oil-Free Foam Propagation*

In this part, the foam propagation is realized as a drainage process in which a non-wetting gas phase is injected into the grain-based pore network fully filled with foaming surfactant solution. A foam lamella will be generated when continuous gas flows across a pore throat being pre-assigned as a foam generation site. Gas is injected into the network at an exact sufficient pressure drop to mobilize a single flowing foam path at the quasistatic term. As shown in Equation (16), the impacts of heterogeneity are studied in this section, which are

controlled by the heterogeneity factor $\sigma$ and grain size range of the commonly used truncated log-normal distribution being adopted to generate grains in the network [50,51].

$$f(R_g, \sigma) = \frac{\sqrt{2}\exp[-\frac{1}{2}(\frac{\ln\frac{R_g}{\overline{R_g}}}{\sigma})^2]}{\overline{R_g}\sqrt{\pi\sigma^2}[\text{erf}\left(\frac{\ln\frac{R_{gmax}}{\overline{R_g}}}{\sqrt{2\sigma^2}}\right) - \text{erf}\left(\frac{\ln\frac{R_{gmin}}{\overline{R_g}}}{\sqrt{2\sigma^2}}\right)]} \tag{16}$$

Then, the radius of the inscribed circle inside three contacting grains is considered as the radius of the pore body, whereas the minimum distance between a pair of adjacent grains is treated as the radius of the pore throat.

(1)  Heterogeneity factor

Three representative heterogeneity factors are applied to study their impact on foam propagation. Figure 14 shows the cumulative grain size distribution generated based on three representative heterogeneity factors, including 0.1, 0.25, and 0.75, representing relatively homogeneous, moderately heterogeneous, and extremely heterogeneous cases. Then, the corresponding frequency distribution of the throat size has been estimated and plotted in Figure 15. Clearly, higher heterogeneity factors will lead to more evenly distributed pore sizes, indicating the formation of more heterogeneous pore networks.

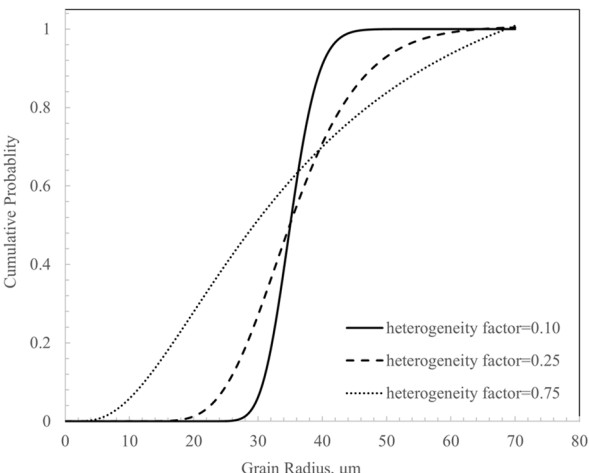

**Figure 14.** Cumulative probability distribution of grain radii at various degrees of heterogeneity.

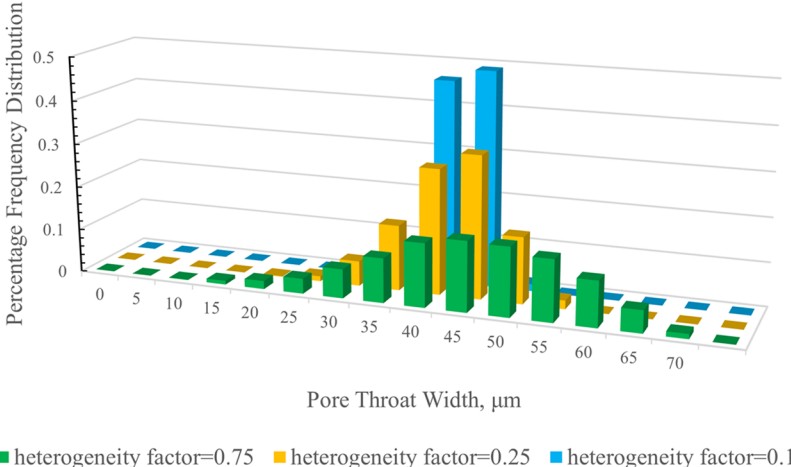

**Figure 15.** Percentage frequency distribution of pore throat width at varying degrees of heterogeneity.

Similar to other immiscible displacement processes simulated based on IPM algorithm, the pressure threshold at the breakthrough of invading phase $p_{BT}'$ and the displacement efficiency $E_D$ are two characteristic indicators distinguishing the impacts of parameters with varying foaming activeness quantified by $p_{SO}$. Two distinct capillarity conditions are discussed, assuming as the weak capillarity condition that foam lamellae only rupture during mobilization, and the strong capillarity condition that static foam lamellae may rupture due to local capillary forces. Benefitting from the features of the PFEN method, the impact of wettability can be incorporated by defining regular ($\theta$ = 120°) and strong drainage ($\theta$ = 170°) conditions.

Under weak capillarity conditions, as shown in Figure 16a, when the heterogeneity factor increases from 0.1 to 0.25, the pressure threshold reduced 61.5% for regular drainage conditions and 68.5% for strong drainage conditions, respectively. However, when heterogeneity factor increases from moderate heterogeneity ($\sigma$ = 0.25) to strong heterogeneity ($\sigma$ = 0.75), the pressure thresholds are reduced for 17.2% and 14.3% for the corresponding wettability condition, respectively. When the capillary condition becomes stronger, as shown in Figure 16b, a less heterogeneous grain distribution still leads to higher breakthrough pressure thresholds when regular drainage conditions are applied. For stronger drainage condition after $\theta$ increases to 170°, the relevance between the heterogeneity factor and breakthrough pressure threshold becomes less significant. Instead, the dynamic distribution of the remaining effective lamellae starts to dominate the nonlinear foam propagation. Similar findings can be drawn from results of displacement efficiency at foam breakthrough, as shown in Figure 17. The results imply that the impact of heterogeneity factor on foam propagation is remarkable when the local capillary force is less significant, allowing stable lamellae presence before mobilization. More static effective lamellae can sustain the relatively stable distribution of resistance contributed by immobile foam banks, whereas the impact from variation of local capillary forces is distinct, which is determined by grain heterogeneity. However, when local capillary forces inside the network are too great to sustain stable foam lamellae, the remaining number of effective lamellae inside the network becomes the more notable factor affecting distribution of resistance exerted from foam lamellae than the capillarity fluctuation caused by heterogeneity. Therefore, as shown in Figures 16 and 17, differences in pressure thresholds and corresponding displacement efficiencies at the breakthrough moment of foam propagation are mainly controlled by other factors related to foam characteristics, such as snap-off probability and lamellae thinning rate, rather than heterogeneity of the grain-based pore network.

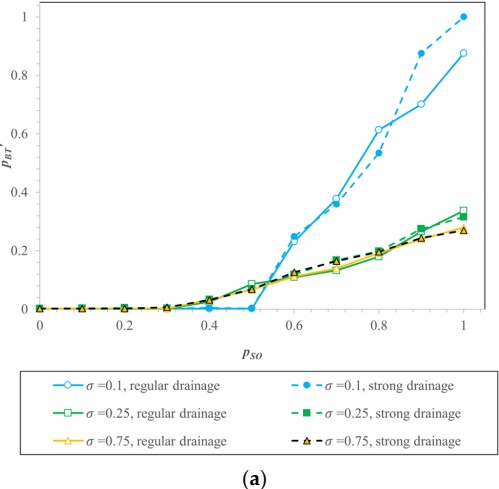

(**a**)

**Figure 16.** *Cont*.

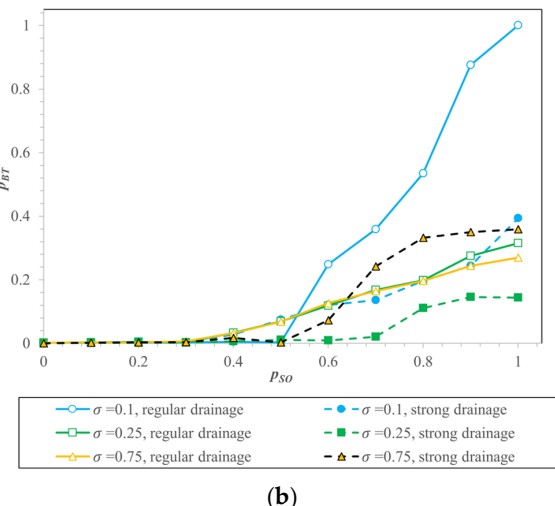

(**b**)

**Figure 16.** The pressure thresholds at the breakthrough with varying foaming activeness based on simulation results. (**a**) weak capillarity condition; (**b**) strong capillary condition.

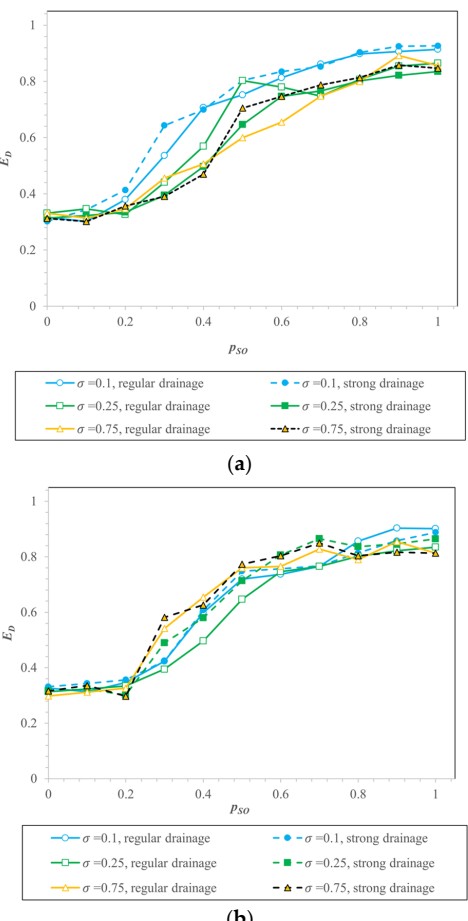

**Figure 17.** The displacement efficiency at the breakthrough with varying foaming activeness based on simulation results. (**a**) weak capillarity condition; (**b**) strong capillary condition.

Figure 18 lists the foam propagation pattern of the invasion processes with varying grain heterogeneity and snap-off probability, which are captured at 75% of the corresponding invasion progress to preserve the features of sufficiently developed invasion front and minimize the scale limits. The results clearly show that the snap-off probability has more remarkable impact on the frontal pattern of foam propagation than grain heterogeneity.

Among the foam propagation patterns shown in Figure 18, active invasion frontal menisci are colored in red, whereas frontal menisci will be excluded from them when a trapped region is formed and relevant frontal menisci are cut off from the outlet of the pore network. From the point of trapping formation, a higher heterogeneity factor and lower snap-off probability will be advantageous in generating a larger trapped region. The heterogeneity factor has a more significant impact while it increases from moderate heterogeneity to higher heterogeneity. For the snap-off probability, as the only model parameter directly determining foaming activeness, it has a more significant impact on foam propagation from a different magnitude while ranging from a low $p_{SO}$ to moderate $p_{SO}$, whereas additional lamellae generation has less contribution to the formation of an immobile foam bank at the corresponding capillarity condition.

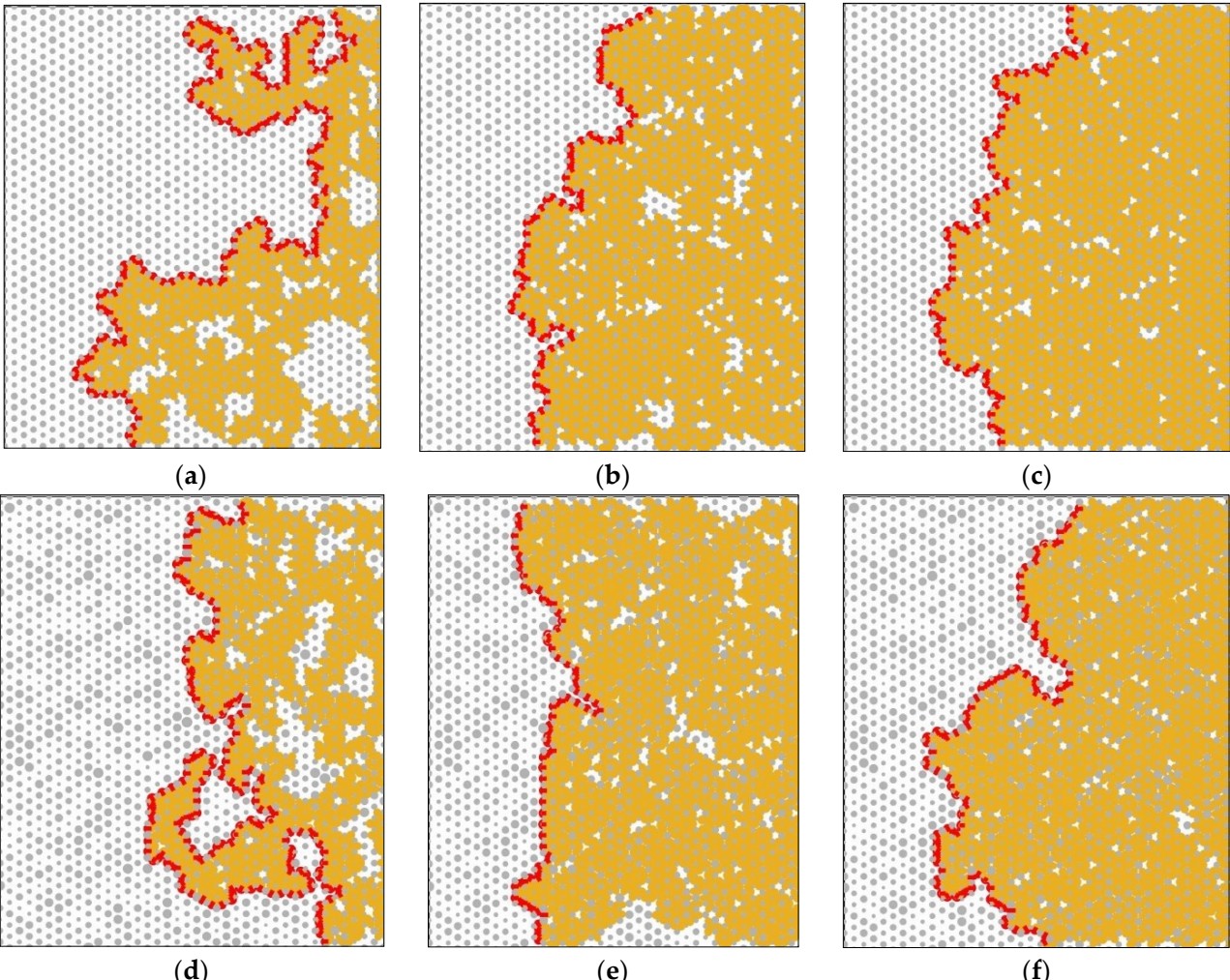

**Figure 18.** *Cont.*

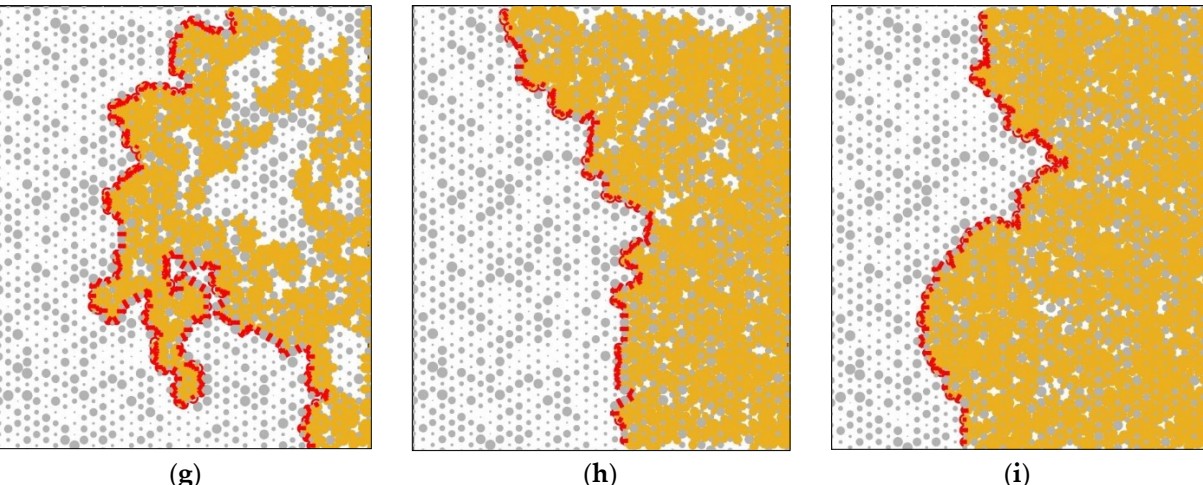

**(g)**             **(h)**             **(i)**

**Figure 18.** Foam propagation patterns with varying grain heterogeneity and lamella generation rate based on simulation results. Invasion phase is colored in yellow, defending phase is in white, and the active invasion candidates and corresponding menisci are colored in red. (**a**) $\sigma = 0.1$, $p_{SO} = 0.3$; (**b**) $\sigma = 0.1$, $p_{SO} = 0.5$; (**c**) $\sigma = 0.1$, $p_{SO} = 0.7$; (**d**) $\sigma = 0.25$, $p_{SO} = 0.3$; (**e**) $\sigma = 0.25$, $p_{SO} = 0.5$; (**f**) $\sigma = 0.25$, $p_{SO} = 0.7$; (**g**) $\sigma = 0.75$, $p_{SO} = 0.3$; (**h**) $\sigma = 0.75$, $p_{SO} = 0.5$; (**i**) $\sigma = 0.75$, $p_{SO} = 0.7$.

Based on the foam propagation patterns listed in Figure 18, the longitudinal active frontal menisci distribution along the direction of gas injection is concluded in Figure 19 to quantify the impact on foam propagation. A more centralized active frontal menisci fraction indicates more a compact, or piston-like, displacement pattern. When $p_{SO}$ is low, such as 0.3, active frontal menisci evenly distribute from the inlet of the network to the invading front, implying the presence of tortuous fingering of the invading phase. For these less active foaming scenarios, the peak of menisci distribution represents the presence of branching points of viscous fingering. The results visualized in Figure 19 indicate that the branching point moves toward the outlet of the pore network as the heterogeneity factor increases. One of the reasons is that the more heterogeneous grain distribution will result in a more complicated fingering mode, including delayed branching of the invading phase. In addition, it tends to form a larger trapped region of the residual defending phase when the heterogeneity factor becomes higher, which also removes a number of menisci from the active frontal list due to current limitations in the trapping identification algorithm. Intuitively, increasing foaming activity will result in more compact propagation patterns, as expected.

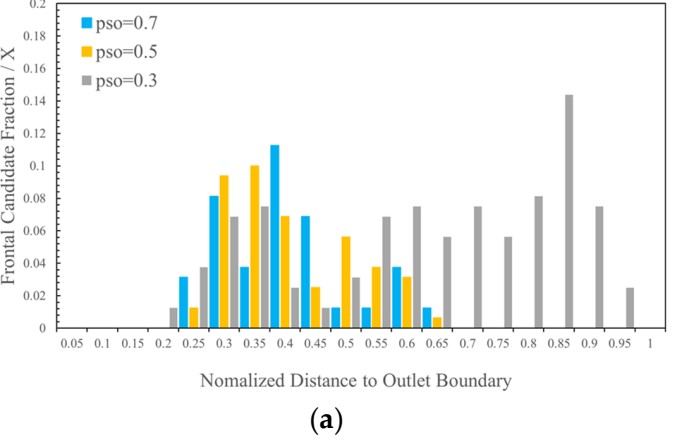

**(a)**

**Figure 19.** *Cont*.

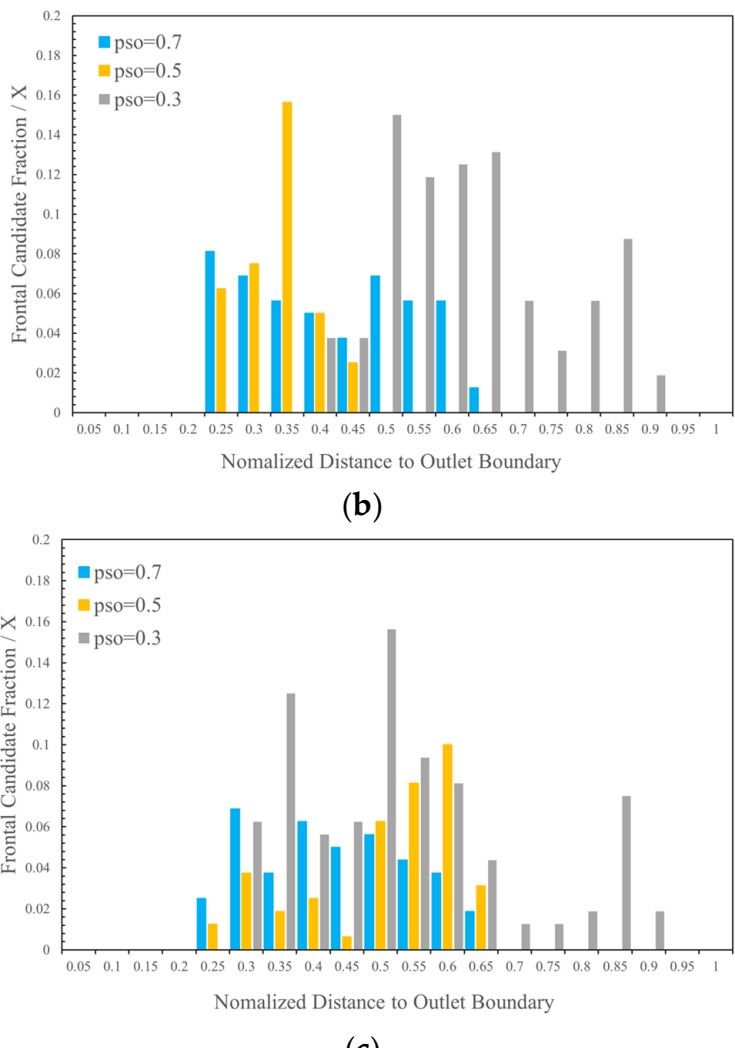

**Figure 19.** Longitudinal active frontal menisci distribution along the direction of foam propagation. (**a**) $\sigma$ = 0.1, plotted based on foam propagation patterns a, b, and c in Figure 18; (**b**) $\sigma$ = 0.25, plotted based on Figure 18d–f; (**c**) $\sigma$ = 0.75, plotted based on Figure 18g–i.

(2)    Grain size range

As one of the parameters used in the truncated log-normal distribution to generate the pore network, the grain size range makes a valuable contribution in providing a more accurate statistical description of the geometric properties of the grain-based pore network. Figure 20 presents the cumulative grain size distribution generated based on three representative grain radius ratios, which is defined as the ratio of the maximum grain radius to the minimum grain radius. Then, the corresponding frequency distribution of the throat size was estimated and plotted in Figure 21 accordingly. Obviously, a narrower range of grain radius selection will result in a more centralized pore throat size distribution, even though the same heterogeneity factor is applied.

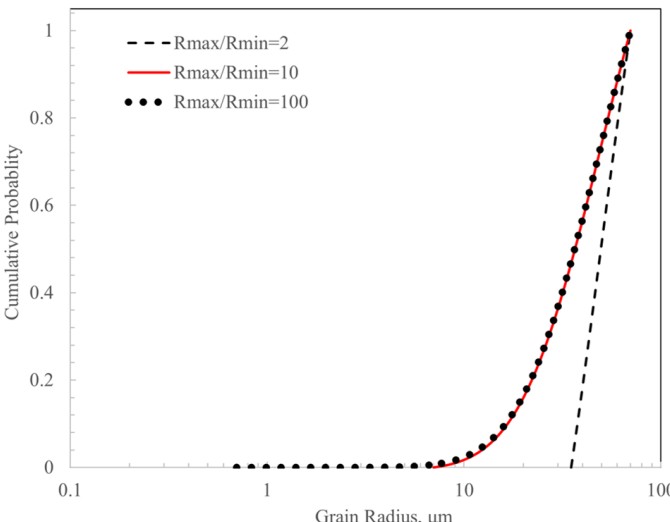

**Figure 20.** Cumulative probability distribution of three representative grain radius ratios. $\sigma = 0.75$.

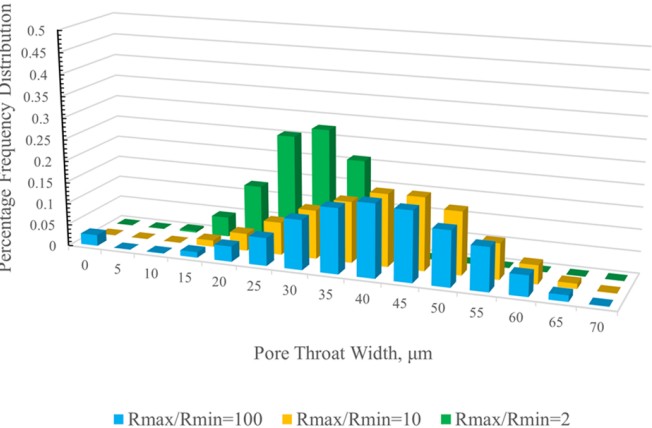

**Figure 21.** Percentage frequency distribution of pore throat width with three typical grain radius ratios. $\sigma = 0.75$.

Figures 22 and 23 present the pressure thresholds $p_{BT}'$ and displacement efficiency $E_D$ at the breakthrough of the invading phase with varying foaming activeness, wettability, and capillarity conditions. For both weak and strong capillary conditions, a decline in the grain radius ratio will lead to an increase in the pressure threshold at foam breakthrough. Although more active foam generation will raise the pressure threshold of foam breakthrough, the additional contribution from higher $p_{SO}$ decreases as well. When static foam lamellae can stabilize themselves at relatively weak capillarity conditions, variation in wettability of grains has distinct impacts on pressure thresholds according to Figure 22a. When the capillarity condition becomes too strong to sustain static lamellae, the impact of wettability is less significant compared with the distribution of the remaining effective lamellae, as shown in Figure 22b. The grain size range does not have a noticeable impact on displacement efficiency at foam breakthrough, either for weak or stronger capillary conditions in Figure 23. This implies that the displacement efficiency is more relevant to the heterogeneity factor than the grain size range, especially when static lamellae are stable.

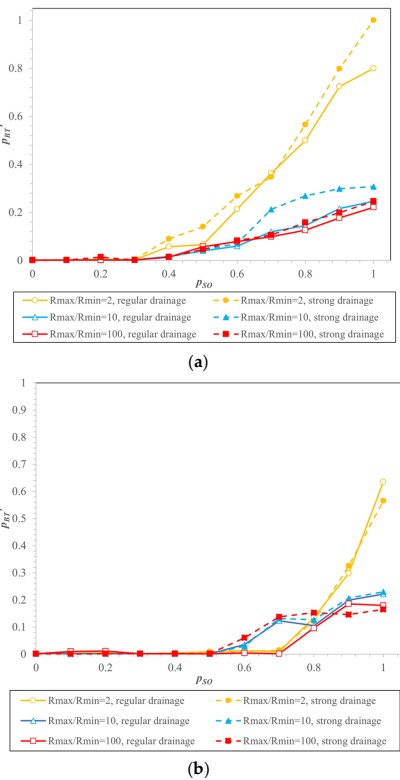

(**a**)

(**b**)

**Figure 22.** The pressure thresholds at the breakthrough with varying foaming activity based on simulation results. (**a**) weak capillarity condition; (**b**) strong capillary condition.

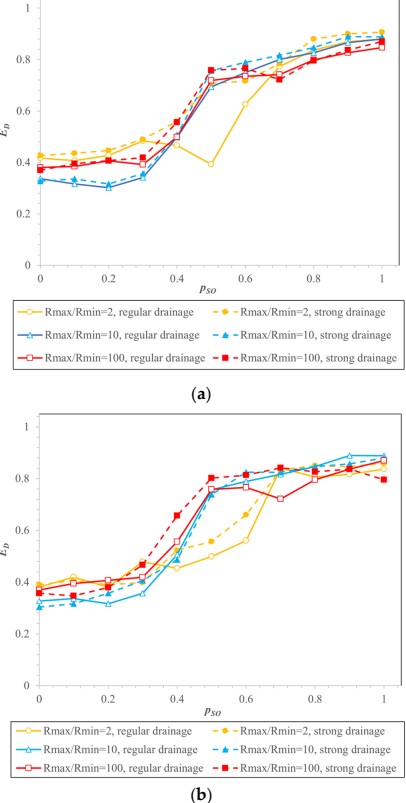

(**a**)

(**b**)

**Figure 23.** The displacement efficiency at the breakthrough with varying foaming activity based on simulation results. (**a**) weak capillarity condition; (**b**) strong capillary condition.

Figure 24 presents a series of foam propagation patterns at 75% of the corresponding total invasion steps of foam breakthrough. Intuitively, foam propagation inside the pore network with a narrower grain size range, such as $R_{max}/R_{min} = 2$, still results in a compact displacement pattern, despite a heterogeneity factor as high as 0.75. When the heterogeneity factor is constant, expansion of the grain size range will lead to more distinct fingering-like displacement features, such as an increasing heterogeneity factor. Similarly, the impact on foam propagation by further expansion of grain size range reduces gradually. The corresponding longitudinal active frontal menisci distribution along the direction of gas injection is concluded in Figure 25 to quantify the impact on foam propagation. Nevertheless, the centralized distribution of active frontal menisci indicates a compact displacing front of foam propagation, whereas evenly distributed menisci imply the presence of viscous channeling. Comparing Figure 25 with Figure 19, reducing the grain radius ratio is more effective in achieving a more compact invasion front than adjusting the heterogeneity factor. During the pore network generation with the truncated log-normal distribution adopted in this work, from either a manufacturing perspective or a statistical perspective, the combination of the grain size range and heterogeneity factor greatly helps generate a customized pore network with a more accurate geometric description.

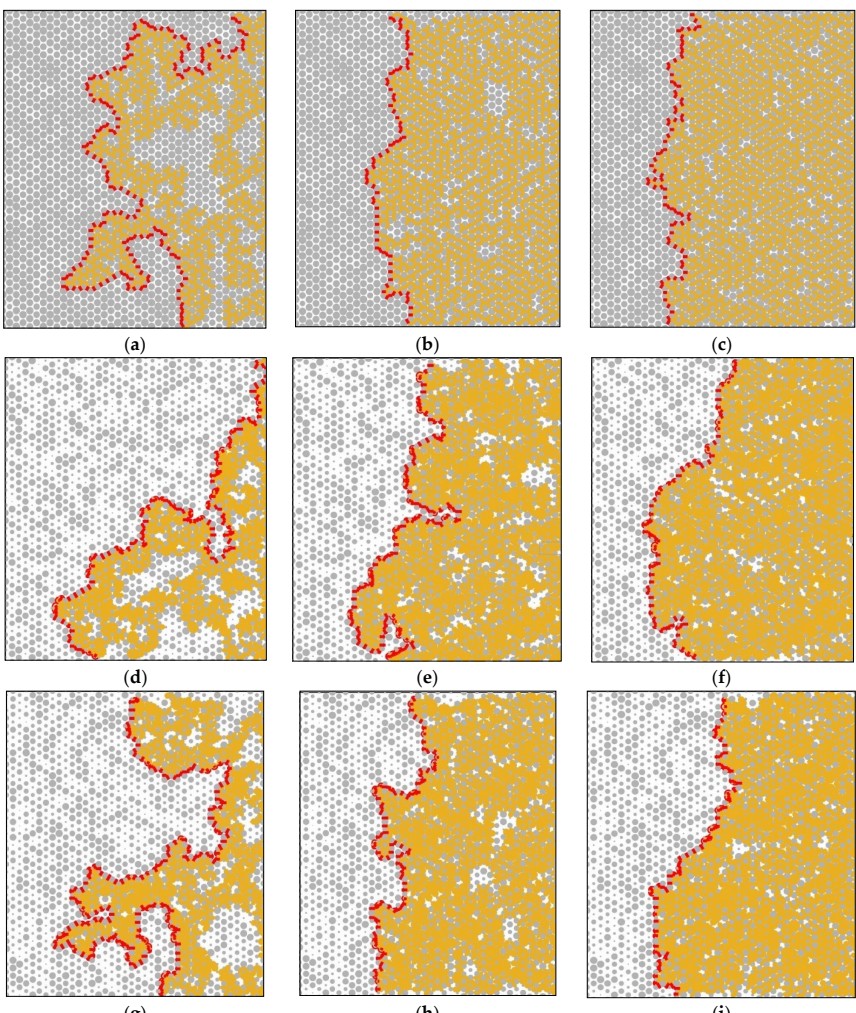

**Figure 24.** Foam propagation patterns with varying grain size ranges and lamella generation rates based on simulation results. (**a**) $R_{max}/R_{min} = 2$, $p_{SO} = 0.3$; (**b**) $R_{max}/R_{min} = 2$, $p_{SO} = 0.5$; (**c**) $R_{max}/R_{min} = 2$, $p_{SO} = 0.7$; (**d**) $R_{max}/R_{min} = 10$, $p_{SO} = 0.3$; (**e**) $R_{max}/R_{min} = 10$, $p_{SO} = 0.5$; (**f**) $R_{max}/R_{min} = 10$, $p_{SO} = 0.7$; (**g**) $R_{max}/R_{min} = 100$, $p_{SO} = 0.3$; (**h**) $R_{max}/R_{min} = 100$, $p_{SO} = 0.5$; (**i**) $\sigma = R_{max}/R_{min} = 100$, $p_{SO} = 0.7$.

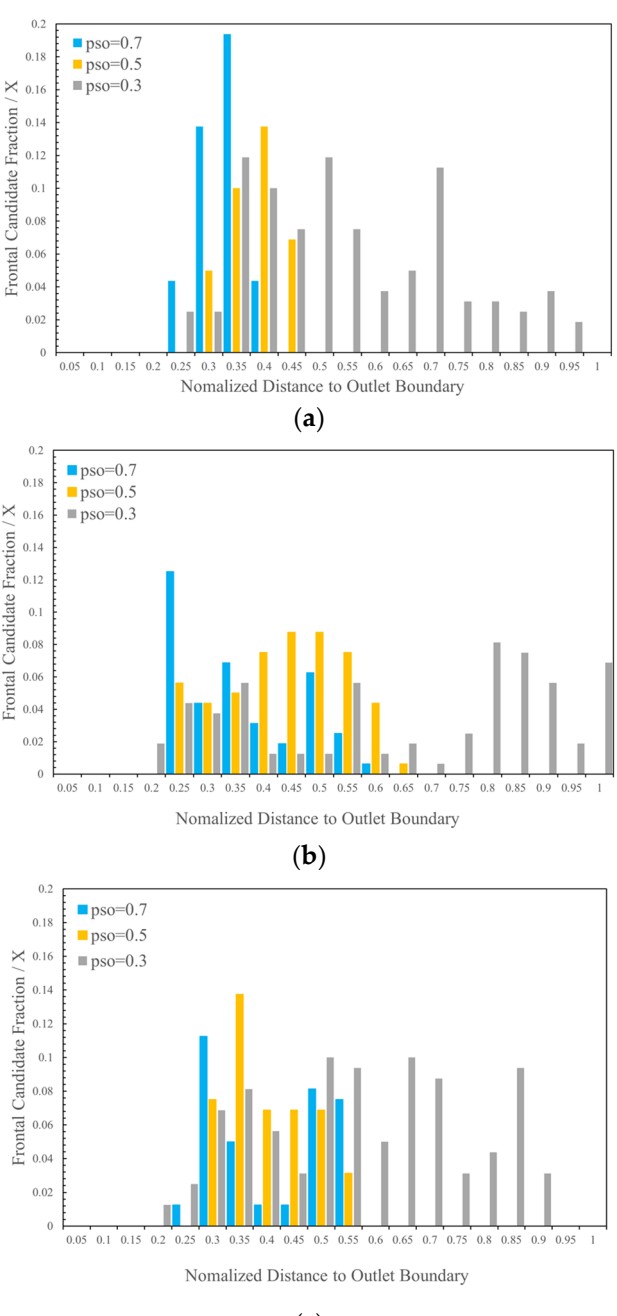

**Figure 25.** Longitudinal active frontal menisci distribution along the direction of foam propagation. (**a**) $R_{max}/R_{min}$ = 2, plotted based on foam propagation patterns a, b, and c in Figure 24; (**b**) $R_{max}/R_{min}$ = 10, plotted based on Figure 24d–f; (**c**) $R_{max}/R_{min}$ = 100, plotted based on Figure 24g–i.

### 3.2. Foam Propagation with Oil-Weakening Effect

Figures 26 and 27 show the cumulative probability distribution of the grain size and corresponding percentage frequency distribution of the pore throat width of this section, respectively. The maximum and minimum grain radii are 70 μm and 0.7 μm, respectively, similar to coarse silt, whereas the heterogeneity factor is set as 0.25 in pore network generation to create a moderately heterogeneous grain-based pore network. As introduced in other sections, two new model parameters are proposed to extend the previous PFEN method from modeling oil-free scenarios into the oil-involved foam propagation processes, including foam quality and minimum continuous segment size.

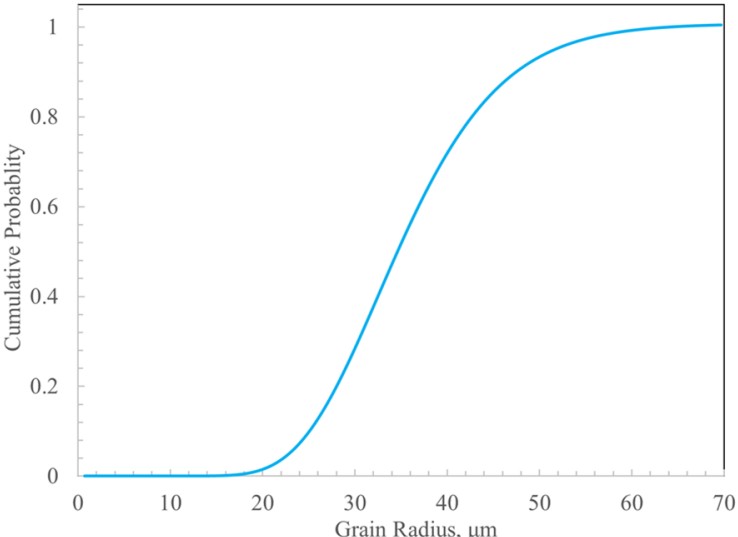

**Figure 26.** Cumulative probability distribution of grain size. $R_{max}/R_{min}$ = 100, $\sigma$ = 0.25.

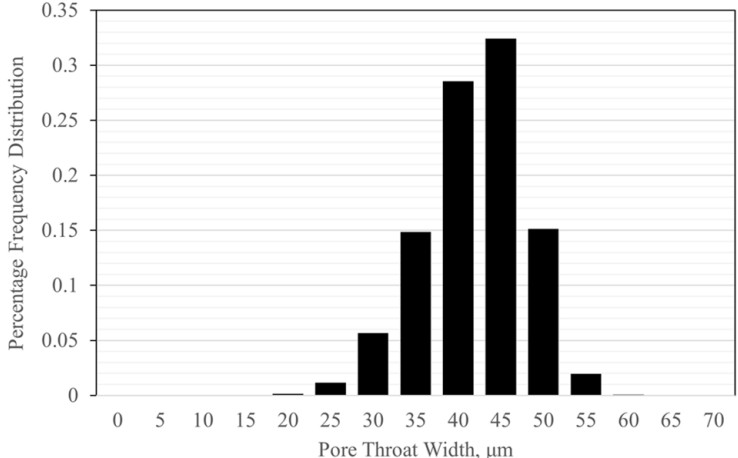

**Figure 27.** Frequency distribution of pore throat width. $R_{max}/R_{min}$ = 100, $\sigma$ = 0.25.

Foam quality is one of the most critical properties of foam propagation inside porous media and directly determines foam regimes. In this work, the foam quality is added into the co-injection foaming process in a stochastic manner, which has an apparent impact on the displacement efficiency and lamellae distribution.

In Figure 28, the displacement efficiency at foam breakthrough with varying foam quality and lamella generation rate are presented. When the minimum segment size is small, implying a dense foam flow with fine texture, the displacement efficiency and foam quality present vague positive correlations in which higher foam quality tends to result in higher displacement efficiency, which shows good agreement with previous studies on the impact of foam quality at multiple scales. Additionally, compared with the liquid phase injected, the additional resistance brought by trapped gas makes a significant contribution to increasing the displacement efficiency.

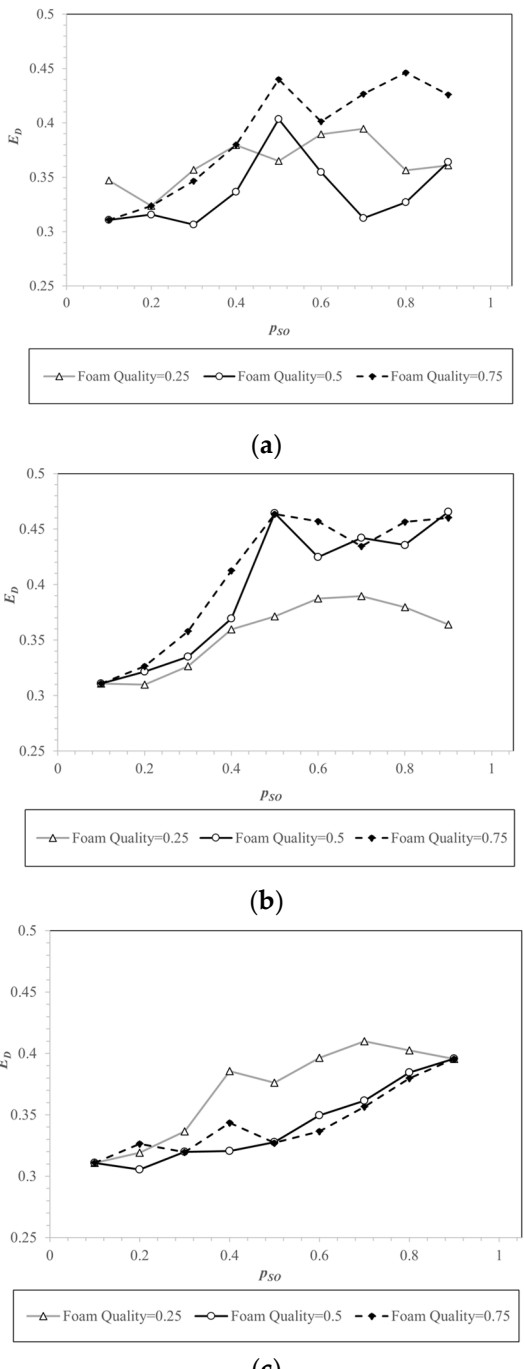

**Figure 28.** Displacement efficiency at foam breakthrough vs. snap-off probability based on simulation results. $R_{max}/R_{min} = 100$, $\sigma = 0.25$. (**a**) $S_{min} = 1$; (**b**) $S_{min} = 10$; (**c**) $S_{min} = 50$.

After rearrangement, the results in Figure 28 are converted to Figure 29 to compare the effect of the minimum segment size of the fluid injected. A reduced minimum segment size of fluid injected can effectively enhance the displacement efficiency as well. Foam quality determines the potential of gas being trapped as foam, and the minimum segment size determines density of foam. In Figure 29a, when foam quality is relatively low at 0.25, the impact of the minimum segment size is negligible on foam propagation inside the pore network, due to the limited presence of gas phase. Based on the results shown in Figure 29b,c, as the foam quality increases, there tends to be an optimized minimum segment size resulting in the highest displacement efficiency at foam breakthrough, such as $S_{min} = 10$ among the results, especially when foam generation becomes more active.

When the minimum segment size is large, such as 50 in Figure 29c, it takes a more active foaming constraint or environment to achieve equivalent displacement efficiency at foam breakthrough.

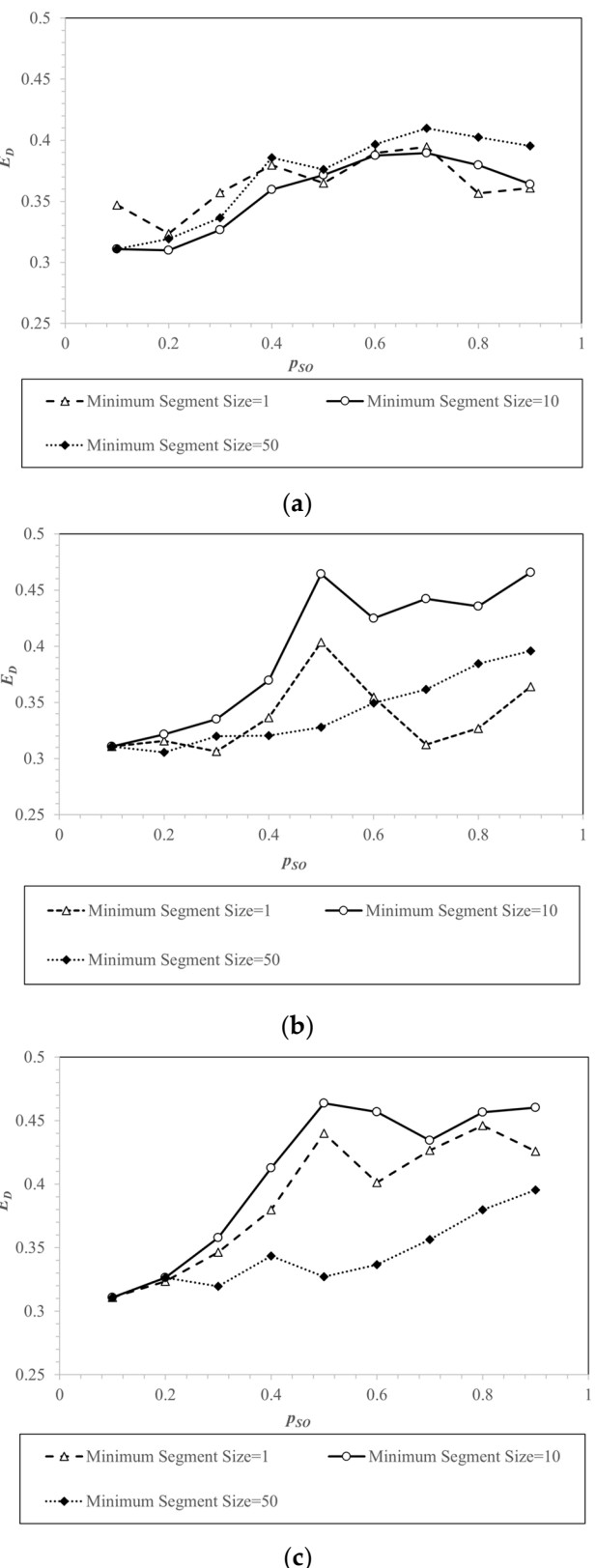

**Figure 29.** Displacement efficiency at foam breakthrough vs. snap-off probability based on simulation results. $R_{max}/R_{min}$ = 100, $\sigma$ = 0.25. (**a**) $\Gamma$ = 0.25; (**b**) $\Gamma$ = 0.5; (**c**) $\Gamma$ = 0.75.

Two groups of foam propagation patterns of co-injection foaming gas and liquid are presented in Figures 30 and 31, which differ in foam quality and foaming activeness, respectively. In these results, pore space filled by red, yellow, green, and blue represents oil, free gas, trapped gas, and liquid phase, respectively. The corresponding lateral fraction and longitudinal distribution of the injected fluids are shown in Figures 32 and 33, respectively. These results quantify the significant changes in lamellae distribution when the overall snap-off probability is reduced from 0.9 to 0.5. After a series of simulations, the results suggest that incremental fraction of the liquid phase within the co-injected fluids not only makes an additional contribution by in situ emulsification to the defense of the oil phase but also prevents potential holdback due to the early coalescence of foam due to the unfavorable oil saturation.

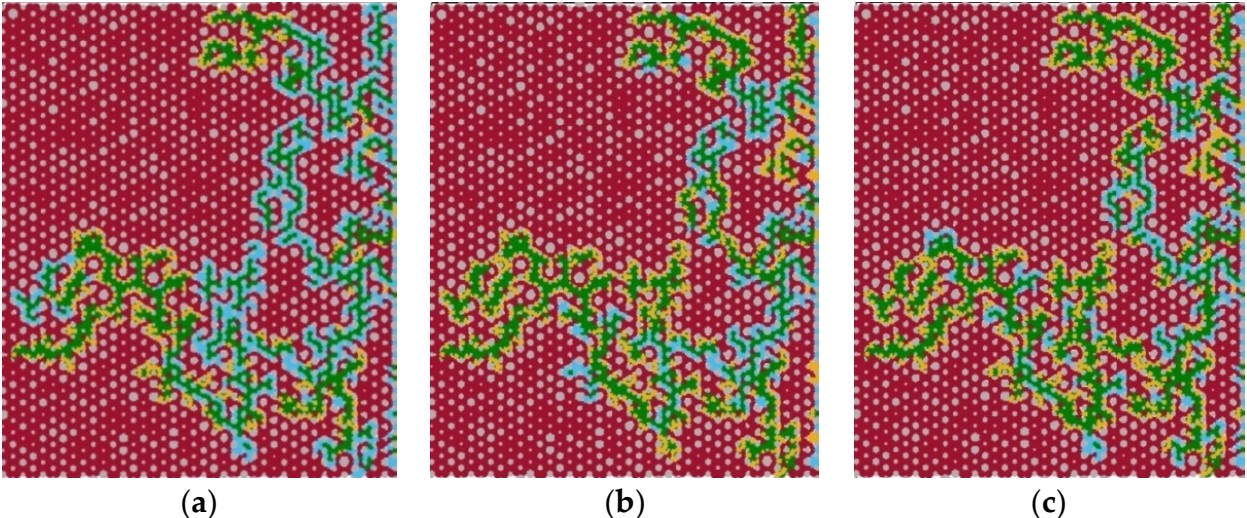

(**a**)  (**b**)  (**c**)

**Figure 30.** Foam propagation patterns with different foam qualities based on simulation results. Defending oil is colored in red, invading liquid is colored in blue, discontinuous gas is colored in green, and continuous gas is colored in yellow. $p_{SO}$ = 0.9, $S_{min}$ = 50. (**a**) $\Gamma$ = 0.25; (**b**) $\Gamma$ = 0.50; (**c**) $\Gamma$ = 0.75.

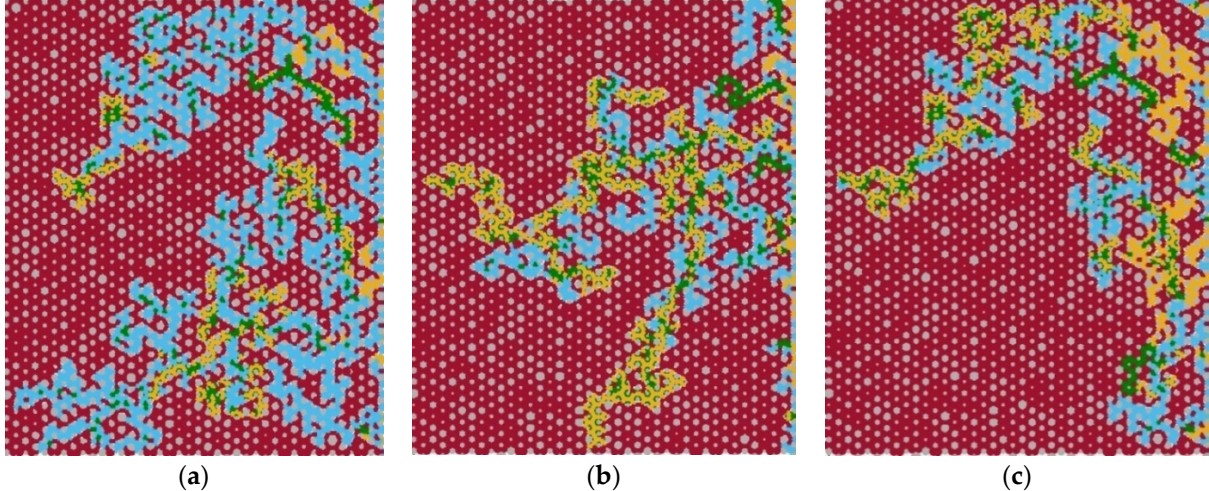

(**a**)  (**b**)  (**c**)

**Figure 31.** Foam propagation patterns with different foam quality. $p_{SO}$ = 0.5, $S_{min}$ = 50. (**a**) $\Gamma$ = 0.25; (**b**) $\Gamma$ = 0.50; (**c**) $\Gamma$ = 0.75.

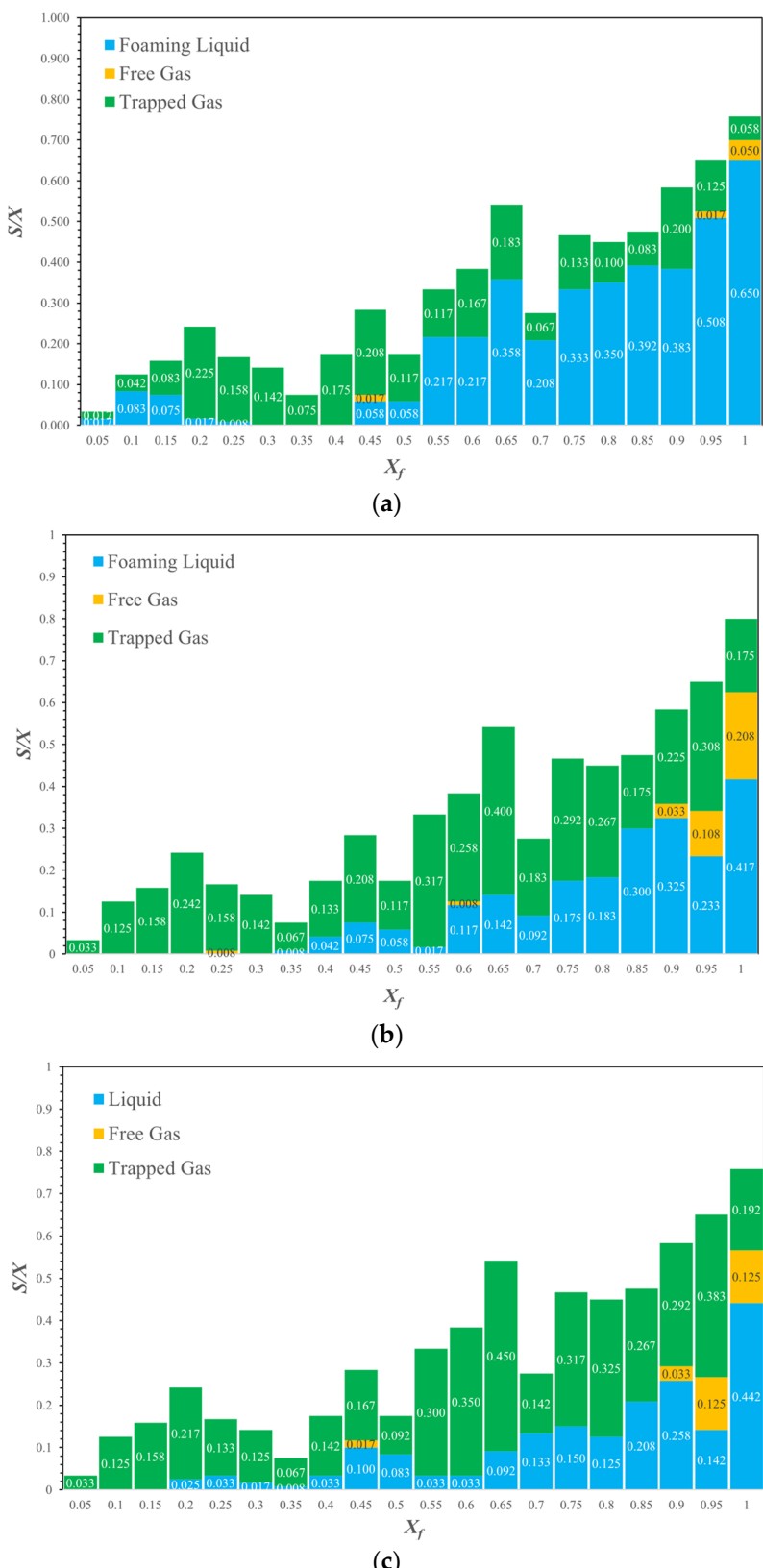

**Figure 32.** Lateral fraction and longitudinal distribution of injected fluids. $p_{SO} = 0.9$, $S_{min} = 50$. (**a**) $\Gamma = 0.25$; (**b**) $\Gamma = 0.50$; (**c**) $\Gamma = 0.75$.

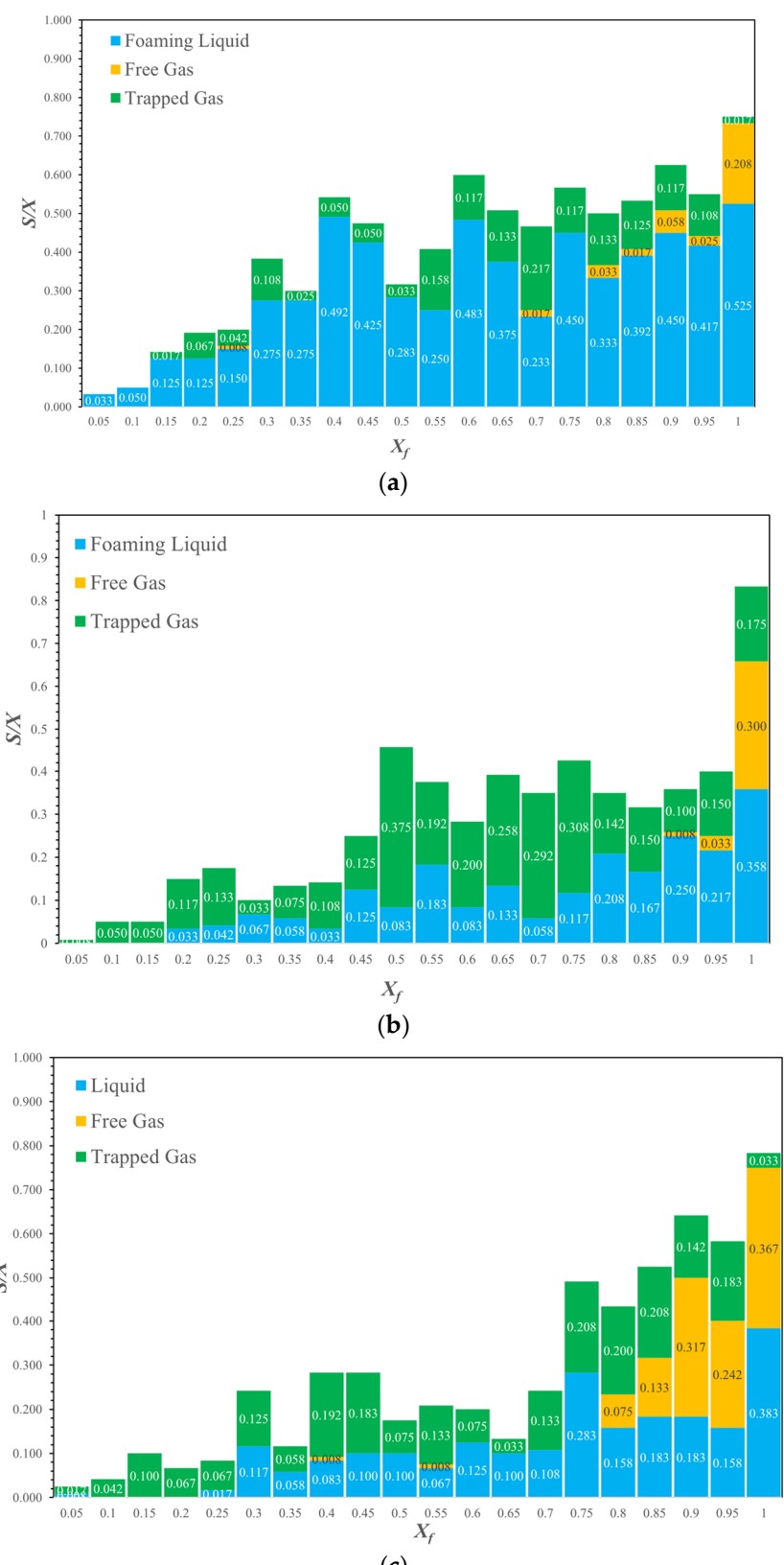

**Figure 33.** Lateral fraction and longitudinal distribution of injected fluids. $p_{SO} = 0.5$, $S_{min} = 50$. (**a**) $\Gamma = 0.25$; (**b**) $\Gamma = 0.50$; (**c**) $\Gamma = 0.75$.

As concluded from foam injection studies on a larger scale, higher foam quality helps create larger immobile foam banks, providing stronger in situ resistance. However, from the

microscale point of view as in this work, such a boost in foam performance is conditional and depends on the foam generation rate, biphasic fluid configuration, oil saturation, and snap-off probability controlled by geometric properties of the rock structure. The formation of the lamellae-rich foam bank, which provides effective mobility adjustment, is dependent on a series of favorable conditions such as high snap-off probability and dense surfactant-alternating-gas configuration, higher foam quality, efficient emulsification, and relatively low remaining oil saturation. Otherwise, additional nonfoaming free gas injected at higher foam quality will play a role in fueling viscous channeling instead of the desired mobility adjustment. Compared with oil-free foam injection scenarios, foaming liquid co-injected with gas not only provides necessary surfactant and liquid film at snap-off sites to ensure stable foam generation but also helps reduce the weakening effect from defending oil by additional emulsification. In conclusion, at the microscale working with interfacial configuration of lamellae, foam quality is still the dominant model parameter that controls multiple key pore-scale mechanisms during the foam propagation processes inside the pore network.

### 3.3. Comparative Experimental Investigation

Currently, although some relevant oil-free or foam-free processes are studied, still lacking are appropriate methods of a similar foaming process (co-injection foam displacing oil) in microscale which can be added and used as a matching group for this proposed algorithmic model. After decades of development, microfluidics has become an efficient and reliable approach to experimentally study pore-scale immiscible displacement processes, such as foam propagation. In this work, microfluidic investigation with the same grain-based pore network as proposed in the modeling section is introduced to provide corresponding qualitative validation. Figure 34a presents the design of a microfluidic chip used in this section, which is a grain-based pore network with a branched highly permeable flowing path inside. The mean grain radius is 48.6 μm, heterogeneity factor is 0.1, and $R_{max}/R_{min}$ is 2. Figure 34b is the experimental apparatus of co-injecting $CO_2$ and foaming liquid (AOS, Bio-Terge AS-40, Stepan) at designated foam quality into the pore network fully filled with silicone oil (General purpose silicone fluids, 100 cp, Brookfield, USA) [52]. Both gas and surfactant are injected at a constant flow rate to ensure a fixed foam quality during the foam propagation, which is 80% in this study. The liquid is injected slowly and smoothly with a syringe pump (Chemyx Fusion 100) at 2 μL/min, whereas the gas is injected at 8 μL/min with a transfer cell controlled by the microfluidic flow controller (Elveflow OB1 MK4, Elveflow MFS) in feedback loop mode.

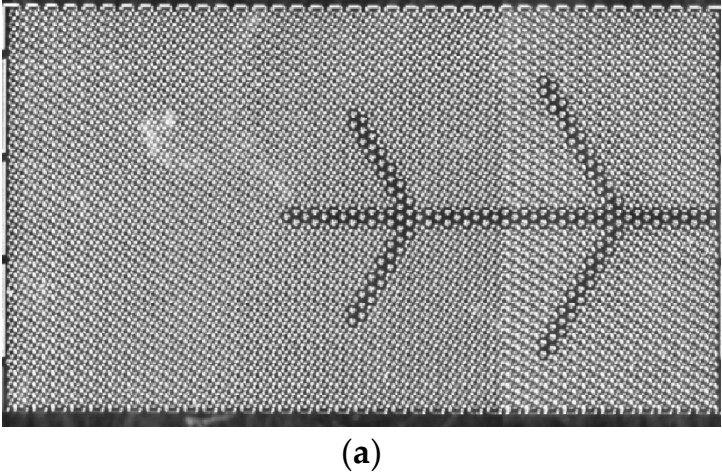

(a)

**Figure 34.** *Cont.*

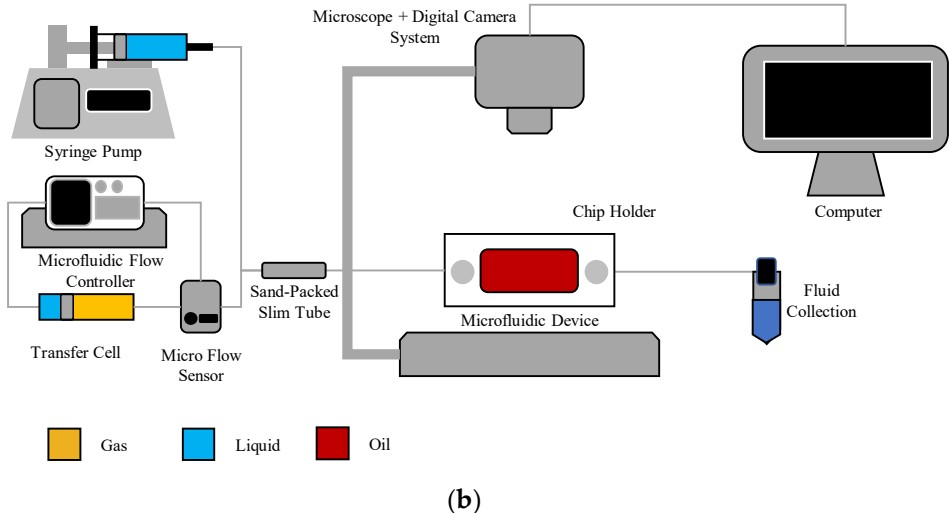

**(b)**

**Figure 34.** Detailed look of the microfluidic chip and experimental apparatus used in the comparative experimental investigation. (**a**) Microfluidic chip with branched highly permeable paths; (**b**) Experimental apparatus of co-injection foaming processes.

Figure 35 presents the foam propagation process inside the microfluidic chip fully saturated with oil, whereas the corresponding simulation results are listed in Figures 36–38, by the sequence of foam propagation patterns, lamellae distributions, and invasion history, respectively. Compared with the experimental results, the simulation results capture the generation features and dynamic foaming phenomena in the same grain-based pore network, from the initial foaming activity to the breakthrough of the invading phase. However, the simulation results underestimate the displacement efficiency due to the limitation in current invasion percolation with memory algorithm which excludes menisci from the candidate list once being excluded from the outlet of the pore network. This simplification is acceptable when modeling a drainage process before the breakthrough occurrence, but it apparently prevents further reduction in the trapped residual defending phase during the post-breakthrough displacement. In Figure 36, the defending oil phase is colored in red, the foaming surfactant is colored in blue, followed by the trapped and free gas phases colored in green and yellow, respectively. To match with the experimental phenomena of successfully making gas phase discontinuous before foam breakthrough, $p_{SO}$ is set as 0.9 in simulation to ensure stable and effective lamellae against significant oil-weakening effects.

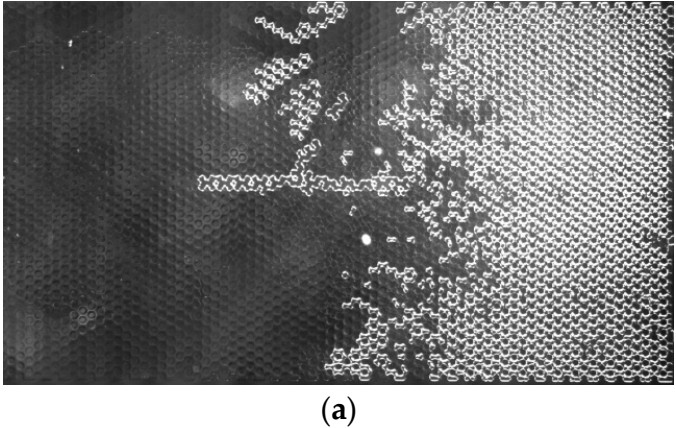

**(a)**

**Figure 35.** *Cont.*

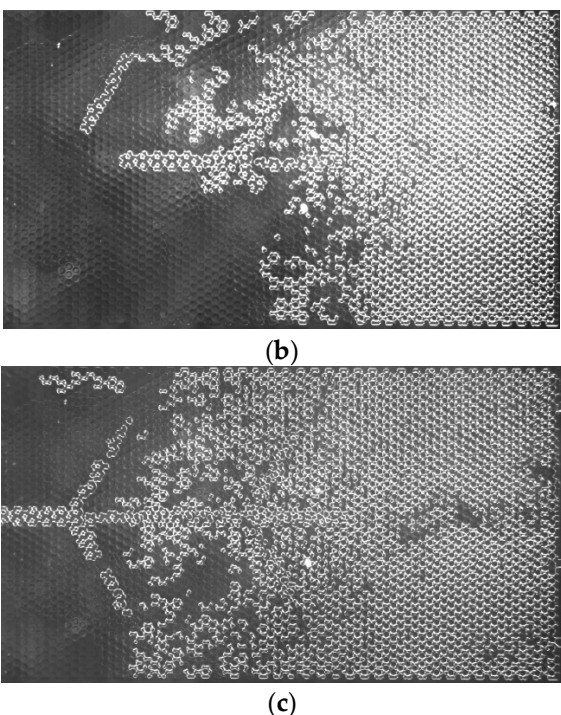

**Figure 35.** Foam propagation in the microfluidic chip with branched highly permeable paths (**a**) $E_D$ = 0.437, PV = 4.575; (**b**) $E_D$ = 0.535, PV = 5.096; (**c**) $E_D$ = 0.719, PV = 7.136.

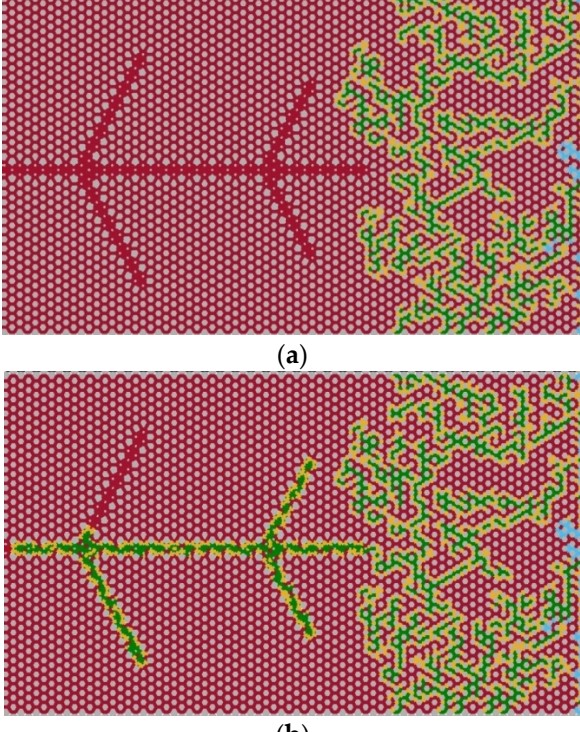

**Figure 36.** Foam propagation pattern during co-injecting processes simulated by the proposed method. (**a**) Multiphase configuration after 1442 steps of invasion simulation; (**b**) Multiphase configuration after 1923 steps of invasion simulation.

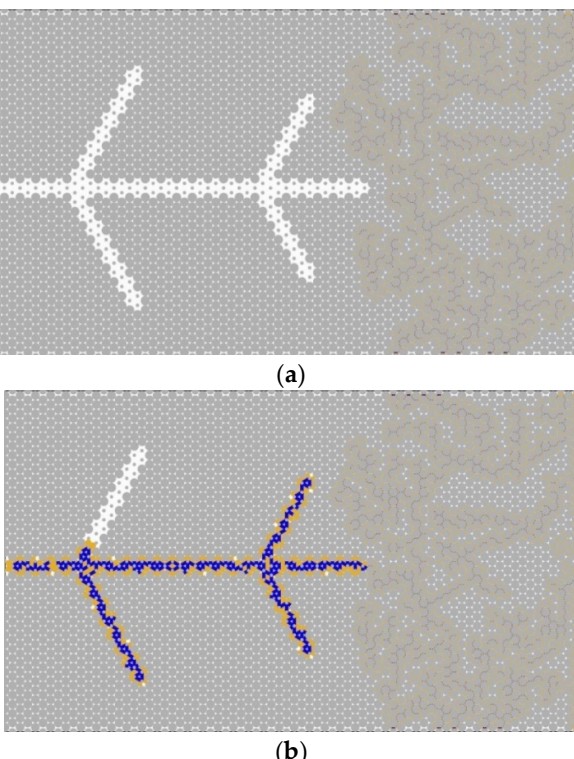

(**a**)

(**b**)

**Figure 37.** Lamellae distribution during co-injecting foaming processes simulated by the proposed method. (**a**) Multiphase configuration after 1442 steps of invasion simulation; (**b**) Multiphase configuration after 1923 steps of invasion simulation.

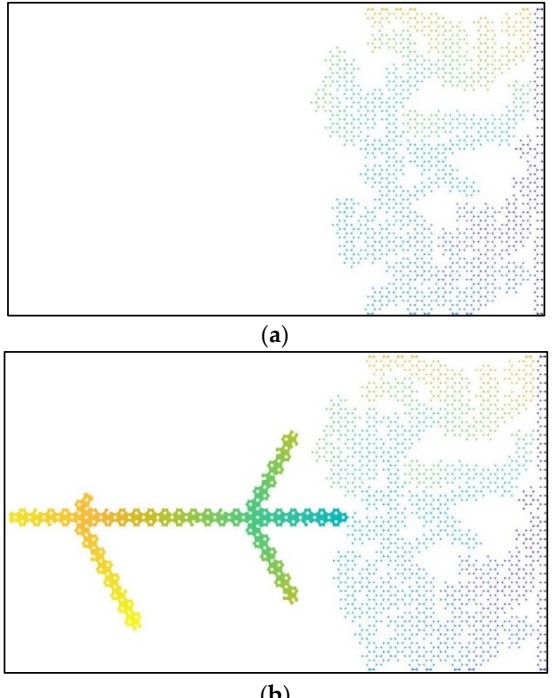

(**a**)

(**b**)

**Figure 38.** Invasion history during co-injecting foaming processes simulated by the proposed method. Yellow end of colormap: the latest invasion step; blue end of colormap: the earliest invasion step. (**a**) Multiphase configuration after 1442 steps of invasion simulation; (**b**) Multiphase configuration after 1923 steps of invasion simulation.

The colormap is used to visualize foam invasion history in Figure 37, which tracks the route of foaming and invading activity inside the grain-based network, which shows that the invasion was not compact and the upper right region of the network was left undisplaced until the foaming gas entered the middle part of the branched region. The defending oil phase prevents compact foam propagation even though an extremely active foaming rate is adopted in the simulation. Additionally, the area of the trapped invasion phase, which is considered as residual oil, is clearly overestimated. A more accurate and efficient trapping identification algorithm is necessary for further development of pore-scale modeling methods for foam propagation.

## 4. Conclusions

With an enhanced approach to quantify interfacial configuration, a novel method is proposed to simulate foam propagation in heterogeneous grain-based pore network with the presence of the defending oil phase. The co-injection process of foaming surfactant and gas is realized in this work with the fully mechanistic model parameters to quantify the foam quality and foam density, incorporated with the oil-weakening effects determined by the quasistatic interactions between oil, liquid, trapped gas, and free gas. Some typical phenomena reported in previous experimental studies can be captured in simulation results, including the effects of heterogeneity factor, grain size range, foam injection strategy, and selective oil weakening. The comparative microfluidic experimental study supports the modeling results in lamellae configurations corresponding to varying foaming activeness, but formation of residual oil is still overestimated due to the limitation in trapping identification of the IPM algorithm. The results suggest that effective wettability adjustment of the rock surface can relieve the negative impacts from unfavorable heterogeneity on the formation of immobile foam banks. Additionally, it supports the core idea of the recently proposed low-tension-gas flooding technology that has foam properties inclining to IFT adjustment ability at relatively lower foam quality, instead of arbitrarily pursuing high-quality strong foam restricted by permeability constraints of tight oil reservoirs with severe heterogeneity.

**Author Contributions:** Conceptualization, J.Y.; Methodology, J.Y. and J.Z.; Software, J.Y., N.L., Z.L., Y.Z., Y.H. and J.Z.; Validation, B.Z. and Y.Z.; Formal analysis, B.Z.; Investigation, N.L., B.Z., Y.Z. and J.Z.; Resources, Y.H.; Data curation, N.L., Z.L. and B.Z.; Writing—original draft, J.Y.; Writing—review & editing, Y.H. and J.Z.; Supervision, J.Z. All authors have read and agreed to the published version of the manuscript.

**Funding:** The authors are grateful for funding from the National Natural Science Foundation of China (Grant No. 12302329, Grant No. 52304020), the Natural Science Foundation of Jiangsu Province (Grant No. BK20230622), the Science & Technology Program of Changzhou (Grant No. CJ20235035), and Jiangsu Key Laboratory of Oil-gas Storage and Transportation Technology (Grant No. CDYQCY202304).

**Data Availability Statement:** Data are contained within the article.

**Conflicts of Interest:** Nu Lu was employed by PetroChina. Zeyu Lin was employed by Dagang Oilfield of China National Petroleum Corporation (CNPC) Ltd. All authors declare that the research was conducted in the absence of any commercial or financial relationships that could be construed as a potential conflict of interest.

## Nomenclature

| | |
|---|---|
| $d_{OA}$ | Distance between the center of grain circle $A$ and the center of arc $A_C B_C$. |
| $p_i$ | Stochastic model parameter controls fluid type injected during a co-injection foaming process. |
| $r_A$ | Radius of grain $A$, µm |
| $r_O$ | Radius of curvature of the meniscus $A_C B_C$, µm |
| $S_{min}$ | The minimum fluid segment size during a co-injection foaming process |
| $\Gamma$ | Foam quality, the gas fraction in foam flow |

| | |
|---|---|
| $\theta$ | Contact angle, ° |
| $\gamma$ | Interfacial tension between invading and defending fluids, mN/m |
| $p$ | Sitewise probability |
| $p_{SO}$ | Snap-off probability |
| $\Pi$ | Disjoining pressure, kPa |
| $\Pi_{max}$ | Maximum disjoining pressure, kPa |
| $tc$ | Time for foam film thinning to critical film thickness, ms |
| $h$ | Foam film thicnkness, nm |
| $\mu_L$ | Viscosity of foaming surfactant solution, mPa·s |
| $R_F$ | Equivalent radius of lamella structure |
| $p_{CA}$ | Local capillary pressure at the constricting part of the grain-based pore space, kPa |
| $t_F$ | Elapsed time of the invasion step, ms |
| $h_{FO}$ | Lamella thickness at the beginning of the corresponding invasion step, nm |
| $h_F$ | Dynamic film thickness of the foam lamella after $t_f$, nm |
| $F_D$ | Coefficient representing additional thinning effect due to oil, nm |
| $V$ | Minimum pressure threshold, kPa |
| $\sigma$ | Heterogeneity factor |
| $R_g$ | Grain size, μm |
| $R_{max}$ | Maximum grain size, μm |
| $R_{min}$ | Minimum grain size, μm |
| $E_D$ | Displacement efficiency, % |
| $p_{BT}{}'$ | Pressure threshold at breakthrough, kPa |
| $X_f$ | Flowing foam fraction |
| $S$ | Saturation, % |
| $X$ | Normalized distance from the outlet boundary |

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
