# Peer review of "Modeling Microscale Foam Propagation in a Heterogeneous Grain-Based Pore Network with the Pore-Filling Event Network Method"

_processes, doi:10.3390/pr11123322_

Round 1

Reviewer 1 Report

Comments and Suggestions for Authors

This paper describes a pore network model for foam propagation in a 2D microfluidic network. It presents a variety of cases within that description and comparison with experiments. However, the paper is unclear on several points, and needs substantial revision for clarity. My major comments are as follows:

1. Page 3: Authors used a different form of Young-Laplace equation in this work compared with common pore network studies. Please state the reason in the methodology section.

2. Page 9: Please explain to readers that how the model parameter p works in determining the foam quality. Also, the typo of injection should be corrected in this line. 

3. Page 9: Further explanation of how the dynamic thickness of foam lamella is incorporated into the quasi-static model should be added.

4. Page 12: In the flow chart and corresponding description, authors should highlight the novel points of this study, which distinguishes it from previous work.

Author Response

Dear Reviewers,

The authors are grateful for your letter and the reviewers' comments on the manuscript (“Modeling Microscale Foam Propagation in Heterogeneous Grain-Based Pore Network with Pore Filling Event Network Method”). All of the editor’s and reviewers’ comments are important and have been considered carefully. The manuscript has been revised accordingly. The response to your kind comments is listed under the section of Reviewer #1 in the attached document.

Reviewer 2 Report

Comments and Suggestions for Authors

Dear authors,

Many thanks for the draft. Please consider the following comments before I can make a final recommendation:

1. In the abstract, please introduce briefly the EOR problem and at the end add some results or conclusions. Moreover, delete "foam" from keywords since this it is too general (or specify a bit more).

2. Figure 1: connect the letters with the subfigures.

3. Add nomenclature table with units.

4. Improve the quality of the images (e.g. Figure 5, 10).

5. Several figures should be explained in more detail, both in the text and captions. For instance, Figures 7, 8, 9.

6. Caption and figures should not be in different pages. Please check this.

7. You are developing a new algorithm; therefore, some validation should be presented or discussed. This part is missing.

8. Unit of degrees is º, please modify.

9. Conclusions should be discussed more thoroughly.

Comments on the Quality of English Language

N/A

Author Response

Dear Reviewers,

The authors are grateful for your letter and the reviewers' comments on the manuscript (“Modeling Microscale Foam Propagation in Heterogeneous Grain-Based Pore Network with Pore Filling Event Network Method”). All of the editor’s and reviewers’ comments are important and have been considered carefully. The manuscript has been revised accordingly. The response to your kind comments is listed under the section of Reviewer #2 in the attached document.

Reviewer 3 Report

Comments and Suggestions for Authors

Comments on the Quality of English Language

There are some grammar errors. Some sentences should be shorten/rewritten.

Author Response

Dear Reviewers,

The authors are grateful for your letter and the reviewers' comments on the manuscript (“Modeling Microscale Foam Propagation in Heterogeneous Grain-Based Pore Network with Pore Filling Event Network Method”). All of the editor’s and reviewers’ comments are important and have been considered carefully. The manuscript has been revised accordingly. The response to your kind comments is listed under the section of Reviewer #3 in the attached document.

Reviewer 4 Report

Comments and Suggestions for Authors

Dear Authors,

I have gone through the manuscript. Its a very detailed and descriptive manuscript. Mathematical model along with experimental investigation is presented for foam propagation in heterogeneous subsurface media. Several experimental setup showing foam propagation are presented supported by the model. 

Paper is very well written, key published references are cited in introduction and literature review. Problem introduction and conclusions are clearly presented. Mathematical models are clear and simulation results provide good demonstration of the results. 

Overall I found manuscript ready for publication without any changes.

Thanks 

Author Response

Dear Reviewers,

The authors are grateful for your letter and the reviewers' comments on the manuscript (“Modeling Microscale Foam Propagation in Heterogeneous Grain-Based Pore Network with Pore Filling Event Network Method”). 

Round 2

Reviewer 2 Report

Comments and Suggestions for Authors

No further comments.